# Ultrathin crown ether-based polyamide membrane for ion-ion separations

Luis Francisco Villalobos [1] ✉, Junwei Zhang [2], Junwoo Lee[2,3], Alex T. Hall[4], Ryan M. DuChanois[5], Camille Violet[2], John Cumings [4], Mingjiang Zhong [2,6] & Menachem Elimelech [5,7]

Membrane-based liquid separations that selectively extract valuable ionic species from water can enhance resource circularity across industries. However, commercial membranes lack the selectivity required to target specific ions. Here we design crown-ether-based polymeric membranes for ion–ion separations using principles inspired by biological ion channels. Ultrathin membranes (~6 nm) are fabricated via interfacial polymerization of crosslinked 18-crown-6 units. The membranes preferentially sorb and transport potassium, which forms the most stable complexes with the crown ether motifs. Sorption experiments show strong potassium preference in mixed-salt environments, where competitive interactions increase selectivity. Transport measurements demonstrate selective permeation of potassium over competing monovalent and divalent cations, with selectivities of ~4 over cesium and lithium. The combination of ultrathin architecture, high crosslinking degree, and high binding-site density enables this behavior. This work establishes interfacial polymerization as a strategy to incorporate macrocycles into membranes for precise ion separation.

Membrane-based liquid separations capable of selectively extracting valuable ionic species from water sources have the potential to enhance circularity across industries, such as metals recovery for clean energy technologies[1,2]. However, state-of-the-art commercial membranes currently lack the solute selectivity required to target specific valuable ions, limiting their effectiveness in such applications. While such membranes separate solutes from water and enable water purification, they fail to selectively recover valuable solutes present in these streams. Therefore, there is a pressing need to develop next-generation membranes capable of differentiating between solutes to selectively extract valuable materials from streams[3,4]. Single-ion selective membranes, which can preferentially permeate a targeted ion while rejecting others of similar charge, hydration shell, and size, are particularly attractive. These membranes could be pivotal in the recovery of valuable ions from waste streams and various applications, including fertilizer recovery from urine, copper recovery in the mining industry, lithium recovery from produced water and brines, and rare earth element recovery from e-waste and ores[1–4]. Single-ion selective membranes also hold potential for applications in batteries and sensors.

Inspired by the selectivity filters of biological ion channels, which exhibit near-perfect ion selectivity and rapid ion transport[5], synthetic ionophores such as crown ethers have been explored to replicate these features in membranes[6]. Crown ethers are a class of macrocyclic ligands that bind to different cations, with their selectivity influenced partly by the match between the cavity size and the target ion's size[7,8]. Crown ether's selective and reversible complexation to ions is a feature that can be leveraged for selective ion transport if they are properly

[1]Mork Family Department of Chemical Engineering and Materials Science, University of Southern California, Los Angeles, CA, USA. [2]Department of Chemical and Environmental Engineering, Yale University, New Haven, CT, USA. [3]Department of Molecular Science and Technology, Ajou University, Suwon-si, Republic of Korea. [4]Department of Materials Science and Engineering, University of Maryland, College Park, MD, USA. [5]Department of Chemical and Biomolecular Engineering, Rice University, Houston, TX, USA. [6]Department of Chemistry, Yale University, New Haven, CT, USA. [7]Department of Civil and Environmental Engineering, Rice University, Houston, TX, USA. ✉e-mail: lf.villalobos@usc.edu

incorporated into a nanochannel, allowing for uninterrupted interactions where ions can reversibly "hop" from one binding site to the next as they traverse through the pore[9–11]. In confined spaces like nanopores, additional factors such as solvation shell rearrangement[12], changes in dielectric properties due to confinement[13], the arrangement of ion-interacting points (including neighboring crown ethers, other functional groups, and fixed charges)[14,15], and crown ether flexibility[16,17], also play important roles. Proof-of-concept experiments, where crown ethers were embedded in lipid bilayer vesicles[15,18–20] or confined within single solid-state nanopores[10], have demonstrated their ability to selectively transport ions in a manner similar to biological ion channels, albeit with less selectivity. In these studies, the ion that selectively complexed with the crown ether building block was transported at faster rates than other ions with the same charge and valence.

Moving from proof-of-concept experiments to functional synthetic membranes that leverage the inherent ability of crown ethers to selectively complex with ions has proven challenging. Only a few complex crystalline channels have successfully harnessed this ability to selectively transport the target ion. For instance, UiO-66 crystals grown inside long and narrow polyethylene terephthalate nanochannels were functionalized with 4′-aminobenzo-15-crown-5 ethers to create artificial sodium channels[21]. In this system, the crown ethers are confined within the pores of the MOF, and despite not all cavities being functionalized, the crystallinity of the MOF provides a somewhat repetitive spacing and order between the crown ether moieties. This structured arrangement enhances the consistency of ion interactions and enables the selective transport of $Na^+$ ions, which form more stable complexes with the 15-crown-5 moieties compared to other ions. The proposed transport mechanism favors $Na^+$ over $K^+$ due to the synergistic effects of size exclusion, charge selectivity, local hydrophobicity, and preferential binding. Another example involves precipitating porous crown-ether crystals inside the ~20 μm tip of a quartz micropipette[22]. A solution containing 1,10-diaza-18-crown-6-ether and $Zn(NO_3)_2 \cdot 6H_2O$ forms a composite upon drying, consisting of 1,10-diaza-18-crown-6-ether-nitrate crystals with densely packed 0.26-nm-wide $Na^+$-selective pores, and ion-impermeable zinc hydroxide nitrate crystals that provide mechanical stability. The interconnected 1,10-diaza-18-crown-6-ether-nitrate crystals create pathways for $Na^+$ ions, which form selective complexes with 1,10-diaza-18-crown-6-ether and permeate faster than other monovalent ions. However, fabricating continuous membranes composed entirely of these crystalline channels remains difficult, as grain boundaries typically dominate ion transport rather than the intended selective pathways.

Unlike proof-of-concept experiments and crystalline channels where ions preferentially interacting with crown ethers exhibited selective transport, the few crown ether-containing polymeric membranes made to date show the opposite effect. In these membranes, ions that bind strongly to crown ethers experience hindered transport compared to others. For example, micrometer-thick polynorbornene copolymer films were designed with distinct blocks: 12-crown-4 ligands for ion selectivity, poly(ethylene oxide) side chains for water regulation, and a crosslinker for structural stability[23]. In the water-swollen matrix, $Na^+$ ions, which bind strongly to the crown ether, show slower transport than $Li^+$. Molecular dynamics simulations confirmed this behavior, attributing it to 12-crown-4's stronger affinity for $Na^+$, which reduces $Na^+$ mobility relative to $Li^+$ in aqueous environments. A separate study synthesized a polymer of intrinsic microporosity incorporating dibenzo-18-crown-6 units, so that every repeating unit contained a potential cation-binding site[24]. The high free-volume linear polymer was processed into micrometer-thick films, and the transport of various monovalent and divalent cations was measured. While the membranes achieved good $Li^+/Mg^{2+}$ selectivity—the target of the study —they did not preferentially permeate $K^+$ ions, despite $K^+$ forming the most stable complexes with the crown ether building blocks used. Instead, higher transport rates were observed for all other monovalent cations tested, indicating a behavior similar to the previous work, where the ion that complexes most strongly with the crown ether units in the membrane exhibited reduced mobility.

In this work, we demonstrate that a crown ether's inherent selectivity can be leveraged in a disordered polymeric system, such as the selective layer of a membrane, to transport the preferred ion selectively. We chose 18-crown-6, which preferentially complexes with $K^+$ ions, as our model crown ether (Fig. 1A). This selection is ideal because $K^+$ lies consistently between $Li^+$ and $Cs^+$ in terms of bare ionic size, hydrated size, hydration energy, and diffusion coefficient in water. While $Li^+$ and $Cs^+$ switch positions depending on the property, $K^+$ remains in the middle across all these parameters (Fig. 1B). Enhanced $K^+$ transport compared to $Li^+$ and $Cs^+$ would highlight the role of 18-crown-6's preferential complexation with $K^+$. To create ultrathin polymeric films from 18-crown-6, we functionalized dibenzo-18-crown-6 with amino groups (Fig. 1C) and reacted it with trimesoyl chloride (TMC) at the aqueous/hexane interface, forming thin crosslinked ionophore films (Fig. 1D). The films were thoroughly characterized using microscopy, scattering, and spectroscopic techniques to unravel their structure and composition. We then tested their selective binding of $K^+$ ions in both single-salt and mixed-salt adsorption experiments. Finally, we performed transport studies to confirm that $K^+$ is selectively transported through the films compared to other ions such as $Li^+$, $Cs^+$, and $Mg^{2+}$.

## Results
### Engineering crown ether-based polymeric membranes
Design guidelines can be derived from our current understanding of how biological ion channels differentiate between ions of the same charge and nearly identical size, selectively transporting the preferred ion thousands of times faster than others[4]. MacKinnon and colleagues' studies on the $K^+$-channel are particularly insightful in this regard[5,25,26]. They found that to transport potassium ions 10,000 times faster than sodium ions, the selectivity filter must: (i) be ultrathin, about 12 Å in the case of the $K^+$-channel; (ii) have a pore size smaller than the hydrated ion but larger than its dehydrated form; and (iii) provide ion-binding sites along the pore walls to stabilize the dehydrated ion. The narrow pore allows only dehydrated ions to pass, and the binding sites favorably interact with $K^+$ ions. The spatial arrangement of these sites stabilizes $K^+$ but not the smaller $Na^+$ ions. Additionally, the close spacing of the binding sites causes adjacent $K^+$ ions to repel each other, lowering the energy barrier for $K^+$ permeation[5,27]. Synthetic membranes that mimic these biological principles—being thin, with sub-nanometer pores capable of dehydrating ions as they pass, functional groups that selectively bind the target ion with moderate binding energy, and optimal spacing between these groups—are promising candidates for achieving similarly high ion selectivity and transport rates[4].

Interfacial polymerization (IP) using crown ether molecules as building blocks (Fig. 1D) offers a promising route to fabricate membranes that align with the key design parameters derived from biological ion channels: being thin, possessing subnanometer pores capable of dehydrating ions as they pass, containing functional groups that selectively bind the target ion with moderate binding energy, and having adequate spacing between these groups. Specifically, first, IP is a self-limiting reaction that enables the fabrication of thin, selective polymer films. Second, by selecting the appropriate crown ether building block, we can introduce functional groups that selectively bind targeted ions with moderate binding energy. Third, the highly crosslinked nature of polymeric structures produced via IP, when using at least one trifunctional monomer, can create membranes with pore sizes small enough to promote partial dehydration of ions during transport. Last, the average spacing between crown ether binding sites

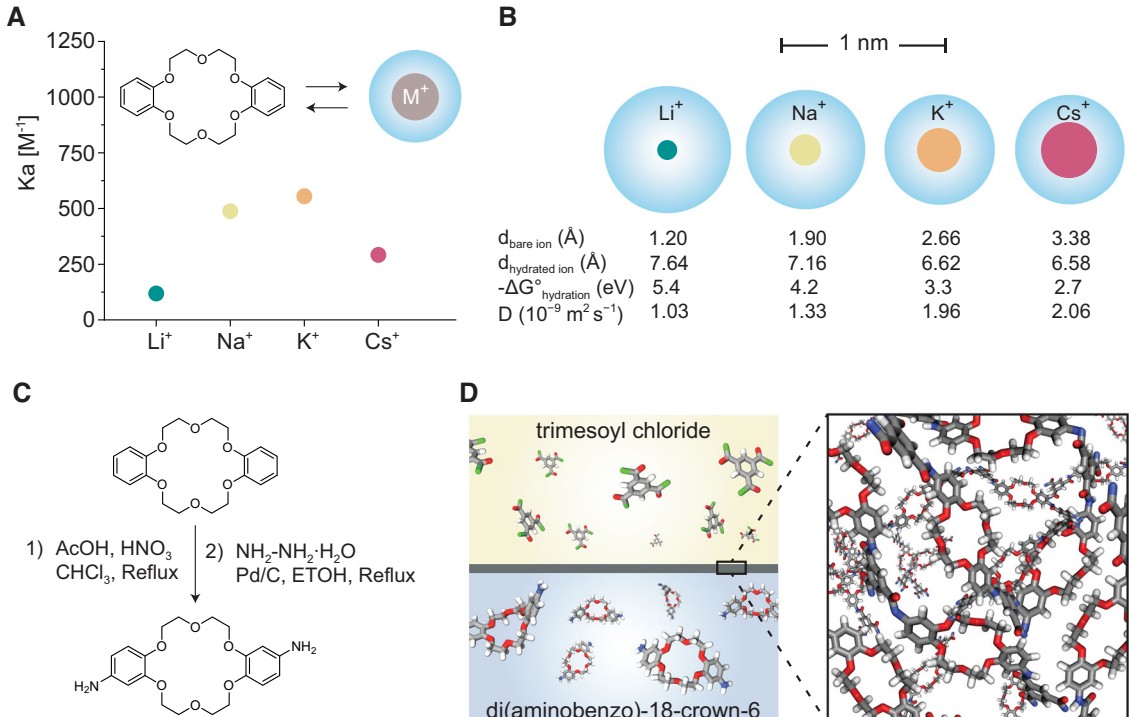

**Fig. 1 | Ion-selective crown ether-based polyamide films via interfacial polymerization. A** Association constant (Ka) between dibenzo-18-crown-6 and monovalent ions (Li⁺, Na⁺, K⁺, and Cs⁺) determined by Dong et al. from ¹H NMR measurements[24]; **B** schematic of four monovalent ions showing their hydrated and bare size at scale; **C** two-step synthesis of trans-di(aminobenzo)-18-crown-6 (DAB18C6) monomer; **D** schematic showing the synthesis of crown ether-based polyamide films via interfacial polymerization of the DAB18C6 (dissolved in the aqueous phase) and the trimesoyl chloride monomer (dissolved in the organic phase).

can be tuned by adjusting the size and connectivity of the acyl chloride monomer in the organic phase.

To effectively incorporate crown ether building blocks into the membrane's selective layer via IP, they must be functionalized with multiple reactive groups—either amine groups to form polyamide layers or hydroxyl groups for polyester layers. This functionalization enables the crown ethers to react with acyl chloride monomers in the organic phase, forming a crosslinked polymer film at the interface of two immiscible phases. IP is the gold-standard technique for fabricating commercial reverse osmosis and nanofiltration membranes, typically relying on small monomers such as m-phenylenediamine or piperazine in the aqueous phase and TMC in the organic phase[28].

The integration of large macrocycles like crown ethers as building blocks introduces challenges due to their limited solubility in the aqueous phase, steric hindrance that can reduce reactivity, and slower diffusion rates to the reaction zone. Despite these challenges, several recent studies have demonstrated the feasibility of incorporating crown ether–based monomers into polymeric thin films via IP, primarily targeting separations such as monovalent–divalent ion pairs[29], proton conduction pathways[30], or pharmaceutical purifications[31]. In these studies, crown ether moieties are largely exploited as macrocyclic cavities that introduce well-defined subnanometer free volume absent in analogous linear or non-macrocyclic polyether monomers, thereby creating additional and more uniform transport pathways that enhance the transport of only certain molecules. Related work has also shown that other bulky macrocycles, including cyclodextrins and trianglamines, can be successfully integrated into ultrathin IP-derived membranes, where their intrinsic porosity similarly enhances solute–solute selectivity[32–34]. Here, we show that the ion-coordination properties of crown ethers can also be harnessed to achieve selective transport between closely related monovalent ions, revealing a distinct transport regime in which selective complexation governs ion–ion selectivity.

The crown ether building block used in this work, trans-di(aminobenzo)-18-crown-6 (DAB18C6), was synthesized in high yield (68% relative to the starting material, dibenzo-18-crown-6) following a previously reported method[35] (Fig. 1C, and Supplementary Fig. 1). A major challenge in using this monomer for IP was its poor water solubility at neutral or basic pH. Solubility was only achievable at acidic pH due to the protonation of the amine groups, though this protonation reduced their reactivity, preventing the formation of high-quality films at pH levels where DAB18C6 was soluble (Supplementary Note 1). This issue was resolved by using a co-solvent: DAB18C6 is readily soluble in DMF, and suitable aqueous solutions for IP were prepared by first dissolving the monomer in DMF, then diluting with water to achieve the target concentration of 0.2% (w/v) in a solvent mixture of 20% DMF and 80% water (v/v). Contacting this solution with a 0.2% (w/v) TMC solution in hexane led to the immediate precipitation of a thin film at the interface. Shaking the vial renewed the interfacial area within fractions of a second, promoting the formation of a gel-like material and further illustrating the rapid reaction rate through visible precipitation. This material was easily recovered by filtration. After thorough washing with water and isopropanol, followed by drying, FTIR analysis confirmed the expected polyamide structure (Fig. 2A). Compared to the DAB18C6 monomer, the resulting material exhibited new peaks at 1650 and 1705 cm⁻¹, corresponding to the carbonyl (C=O) stretching vibrations of amide and carboxylic acid groups, respectively. Additionally, the characteristic asymmetric and symmetric N–H stretching peaks of the primary amine groups in the DAB18C6 monomer disappeared after polymerization, replaced by a broad peak in the 3000–3600 cm⁻¹ region in the polyamide film spectrum, likely due to absorbed moisture.

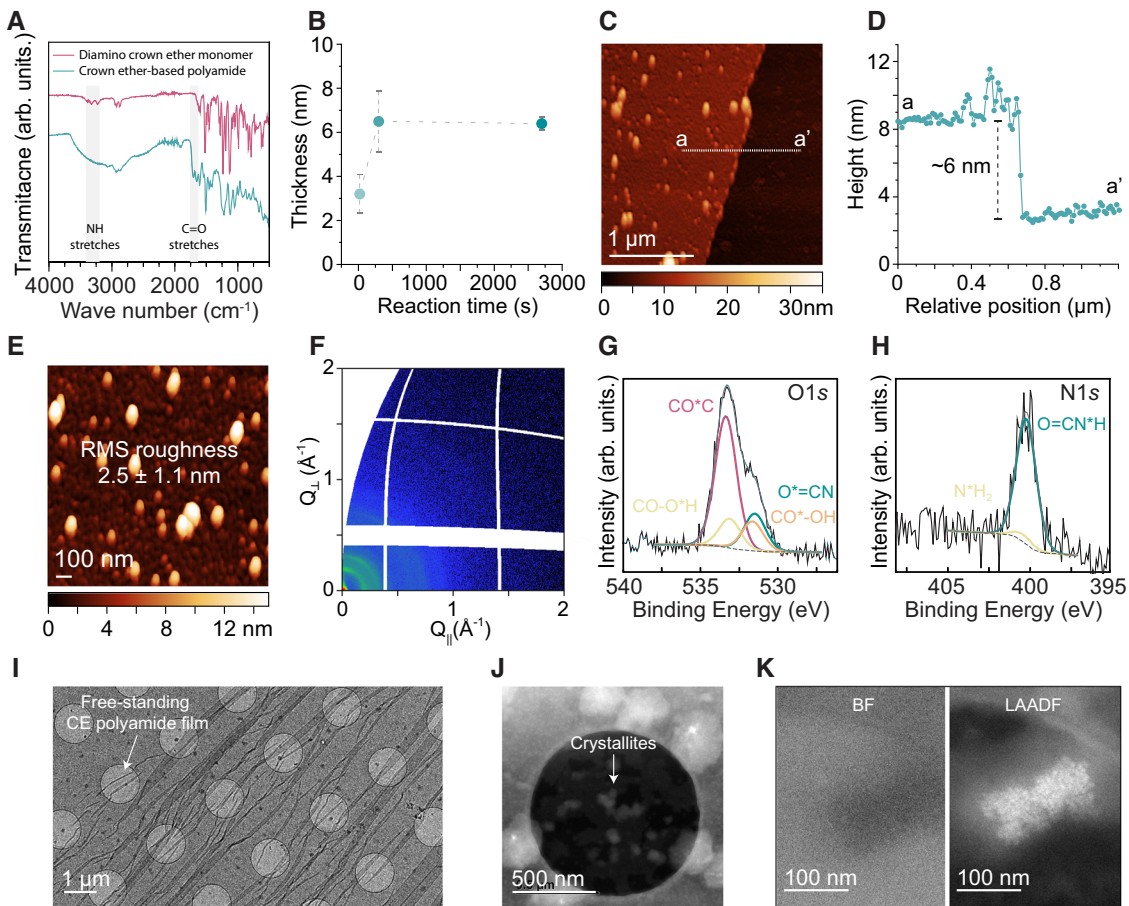

**Fig. 2 | Characterization of the crown ether-based polyamide film made with interfacial polymerization. A** FTIR of DAB18C6 before and after polymerizing it with TMC. **B** Crown ether-based polyamide film thickness as a function of reaction time. Films were transferred to silicon wafers, washed with water, dried at room temperature, and scratched with a needle before AFM step-height analysis. **C** AFM image of a crown ether-based polyamide film prepared using 5 min reaction time and its corresponding cross-sectional profile (**D**). **E** High magnification AFM image of the crown ether-based polyamide. Comprehensive characterization using GIWAXS two-dimensional image (**F**) and XPS (**G, H**), revealing an amorphous polyamide matrix. **I** Low magnification cryogenic TEM image showing a crown ether-based polyamide film covering the ~1 μm holes of the holey carbon support. **J** Cryogenic LAADF-STEM image revealing the presence of crystallites in the polyamide matrix. **K** Cryogenic BF and LAADF-STEM images of a characteristic polycrystalline aggregate at high magnification. All membranes were synthesized from 0.2 wt.% solutions of TMC in hexane and DAB18C6 in DMF-water (20/80 v/v), with a standard reaction time of 5 min unless otherwise noted. Data are presented as mean ± standard deviation (SD) from $n = 3$ independent measurements (**B**).

The thinness of crown ether-containing membranes is crucial when aiming to leverage the reversible and selective interactions of crown ether groups for ion transport. In such membranes, where selective transport relies on these reversible interactions, we hypothesize that the dominant transport resistance for the target ion arises from the movement through the network of binding sites[4,36]. This process is governed by the repeated reversible binding and release of the ion as it traverses the membrane. A thicker membrane is therefore expected to slow this process, as the ion would need to undergo more interactions before crossing to the other side. Conversely, for competing ions that interact weakly with the binding sites, transport is not selectively facilitated and may be governed by interfacial partitioning into the membrane, such that increasing thickness has a comparatively smaller effect on their transport[36,37]. As membrane thickness increases, the transport of the target ion—selectively binding with the functional groups—can be penalized more strongly than that of competing ions. We recently demonstrated this using a model system containing a high density of Cu²⁺-selective iminodiacetate groups, where membrane thickness could be precisely controlled, and increasing thickness led to a decline in copper selectivity[36].

The selected reaction conditions (0.2 w/v% of each monomer, with a solvent mixture of 20% DMF and 80% water (v/v) for the aqueous phase, and hexane for the organic phase) led to the synthesis of ultrathin films. The self-limiting nature of IP was evident, as the membranes reached a maximum thickness of approximately 6 nm, even with extended reaction times (Fig. 2B). Since a 5-min reaction was sufficient to achieve this thickness (Fig. 2C, D), we conducted all subsequent studies using a 5-min reaction time. This approach consistently produced smooth films with a low surface roughness of only 2.4 nm (Fig. 2E) and a moderately hydrophilic surface, as indicated by a water contact angle of 67 ± 3°. The thinness of these films is well-suited to leverage the selective complexation of 18-crown-6, minimizing the number of binding events required for ion permeation across the membrane.

The appropriate binding energy—neither too strong nor too weak—and the spatial arrangement of crown ethers within the selective layer are also critical for effective ion transport[14]. Strong, irreversible complexation may promote selective partitioning but can trap the target ions, allowing competing ions to permeate faster despite their lower partitioning. Notably, most crown ethers form reversible complexes with ions, and the stability of these complexes is ion-specific[38]. Studies

embedding synthetic crown ether channels into lipid bilayer vesicles have shown that ion binding preferences dictate selective ion transport and ion transport via ion hopping from one crown ether site to the next[15,18–20]. Channels composed of 21-crown-7 molecules preferentially transported Cs[+] ions, while those made of 18-crown-6 or 15-crown-5 selectively transported K[+] and Na[+] ions, respectively[18].

Accordingly, dibenzo-18-crown-6 was selected as the membrane-forming building block because its cavity size and donor functionality are well matched to K[+] and are known to retain preferential K[+] coordination even when conformational freedom compared to 18-crown-6 is reduced. Its cavity size and functional groups allow it to form stable, selective complexes with K[+], as shown in Fig. 1A, which is based on data from Dong et al.[24]. Through proton NMR titration, Dong et al. monitored the chemical shift changes in dibenzo-18-crown-6 as it interacted with various metal ions. The binding strength difference was significant for Cs[+] and Li[+] compared to K[+], whereas the preference for K[+] over Na[+] was relatively minor.

In the prepared ultrathin films, selective complexation with K[+] ions is anticipated. Crown ethers free in solution can adapt their conformation to optimize coordination with a given ion; however, when incorporated into a densely crosslinked polyamide network, this flexibility is constrained, and the idealized cavity geometry may be distorted. As a result, ion coordination within the membrane should not be viewed as occurring through discrete pores of fixed size, but rather through locally constrained coordination environments defined by the crown ether moieties and the surrounding polymer matrix. These constrained sites likely exhibit reduced selectivity compared to free crown ethers in solution, consistent with prior observations of crown ethers under restricted conformational freedom[15].

The optimal spacing between crown ethers, where effective ion hopping occurs, has been identified to be within 6–11 Å in vesicle-type studies[18]. Ottis et al. observed that a 6 Å spacing was ideal for Na[+] ion hopping, with effective transport up to 11 Å[18]. These distances are comparable to those in biological ion channels, such as the 11.6 Å between ion-relay sites in gramicidin A[39] and the 7.5 Å between K[+] ions in the KcsA K[+] channel[5]. In other systems, such as the one proposed here, the optimal distance may differ due to factors like confinement and the chemical environment, but it is likely to be within a similar range. Overall, the spacing between crown ethers should be sufficiently short to promote ion-ion repulsion but not so short that it encourages less selective, sandwich-type complexes. When crown ethers are spaced closer than ~5 Å, these sandwich-like complexes form, disrupting intrinsic selectivity[15]. For example, while 15-crown-5 generally favors Na[+] ions, the formation of such complexes shifts its selectivity towards larger K[+] ions[15,19].

Confining bulky molecules such as crown ethers into a 6 nm space could force some order in the structure, as has been previously observed for ultrathin polyamide membranes made with, albeit bulkier, cyclodextrin building blocks[33,40]. To investigate whether similar ordering occurred in our films, we performed grazing-incidence wide-angle X-ray scattering (GIWAXS). The GIWAXS data revealed only a broad, diffuse scattering pattern without distinct Bragg peaks, indicative of an amorphous arrangement rather than any ordered crystalline or semi-crystalline structure (Fig. 2F, and Supplementary Fig. 3). This suggests that, despite the ultrathin configuration and the presence of bulky crown ether groups, the polyamide chains remain randomly oriented in the membrane, confirming the amorphous nature of the material.

While the average distance between crown ether units could not be directly measured, a simple structural estimation suggests it is likely within an attractive range for ion hopping. Assuming a perfectly flat and linear structure, the maximum separation between two crown ether centers in the polyamide would be approximately 1.5 nm. However, considering the random and crosslinked nature of the structure, along with the flexibility of the amide bonds, the actual spacing in the

3D structure is likely less than 1 nm, which would be conducive to ion hopping (Supplementary Fig. 4). In our prepared membranes, crown ether molecules are arranged randomly, and while aligning them could improve ion hopping efficiency[20,41], it is not necessary for selective ion transport. For example, Siwy and coworkers randomly attached crown ethers to the walls of a sub-2 nm solid-state pore and achieved K[+]/Na[+] selectivities close to 80[10], demonstrating that an ordered assembly is not a prerequisite for successful ion transport.

## K[+] selective polyamide thin film made from crosslinked 18-Crown-6 building blocks

The formation of a crosslinked polyamide network made of crown ether moieties was confirmed through XPS characterization. Thin polyamide films were transferred to gold-coated substrates for XPS analysis, providing detailed insights into the bonding environments within the membrane. The O1s core-level spectrum displayed distinct binding energies indicative of C–O–C bonds from the crown ether structure, along with amide linkages, confirming the formation of the intended crosslinked network (Fig. 2G). Notably, the spectrum revealed both amide and carboxylic acid groups in nearly equivalent concentrations. This high concentration of carboxylic acid groups, resulting from the hydrolysis of unreacted acyl chloride groups from TMC upon contact with water in the aqueous phase, aligns with the slower reaction kinetics expected for the bulky crown ether monomer. The slower diffusion rates and pronounced steric hindrance of this larger monomer limit amine availability at the reaction interface, thereby increasing susceptibility to acyl chloride hydrolysis. By contrast, smaller monomers commonly used in IP, such as m-phenylenediamine and piperazine, react more rapidly due to their compact size and higher mobility, thus minimizing acyl chloride hydrolysis during membrane formation. The N1s core-level spectrum provides additional evidence of successful crosslinking of the crown ether monomers, showing a single dominant nitrogen environment corresponding to amide nitrogen formed by reaction of acyl chloride with amine groups (Fig. 2H).

Cryogenic scanning and transmission electron microscopy (S/TEM) analysis of the freestanding 6-nm-thick crown ether-based polyamide films demonstrated their mechanical robustness as they spanned micrometer-sized openings in Quantifoil grids (Fig. 2I). Moreover, the films displayed high flexibility, with no signs of cracking, even in areas with visible folds. Cryogenic low-angle annular dark-field (LAADF) STEM imaging revealed evenly distributed crystallites throughout the films (Fig. 2J and Supplementary Fig. 5), which appeared as aggregated nanoparticle-like structures upon closer inspection (Fig. 2K). Tilting the sample during imaging produced contrast fluctuations characteristic of polycrystalline regions (Supplementary Fig. 6). LAADF-STEM was crucial in identifying these crystallites, as they appeared as indistinct blobs in bright-field imaging (BF) (Fig. 2K, left). These crystallites are likely precipitates of the DAB18C6 monomer, which has poor water solubility. While the majority of the membrane was amorphous, the crystallites were consistently dispersed throughout the structure.

Ion sorption experiments revealed a distinct preference for K[+] over Cs[+] in the crown ether-based polyamide films (Fig. 3), despite both ions having nearly identical hydrated sizes (Fig. 1A). The experiments were conducted at low pH to ensure protonation of all carboxylic acid groups within the membrane[42], thereby minimizing their contribution to ion adsorption. Real-time quartz crystal microbalance (QCM) tracking showed significant mass changes upon exposure to 0.5 M solutions of CsCl and KCl (Fig. 3B), with frequency decreases corresponding to ion uptake. The calculated molar areal uptake from these single ion sorption experiments confirmed a higher affinity for K[+] (Fig. 3C), though this affinity was less pronounced than anticipated based on the large difference in equilibrium constants for the crown ether building block when free in solution (Fig. 1B). Selectivity tests

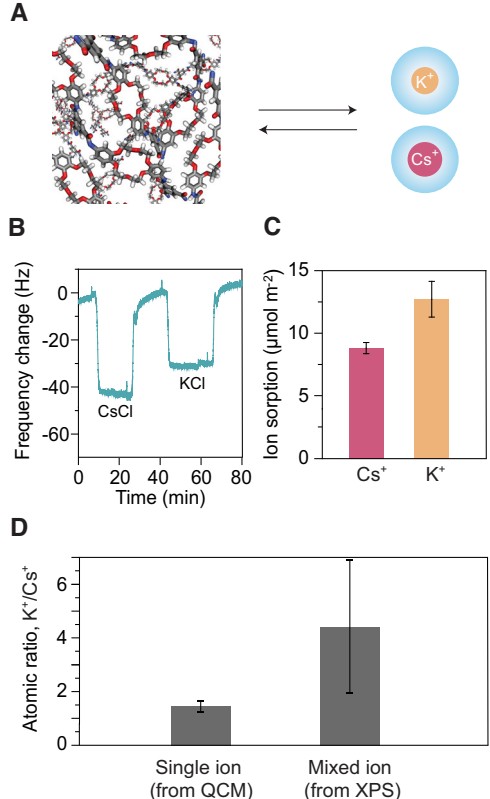

**Fig. 3 | Monovalent ion sorption by the crown ether-based polyamide film.**
**A** Illustration of a crosslinked crown ether-based polyamide matrix adsorbing and
desorbing $Cs^+$ and $K^+$ ions. **B** Real-time, in-situ tracking of CsCl and KCl partitioning
in the polyamide film via QCM, where frequency shifts indicate mass changes upon
exposure to 0.5 M solutions of CsCl and KCl (pH = 3.2). **C** Calculated average molar
areal ion uptake from QCM data. **D** Ion sorption selectivity determined from single
salt (using QCM) and mixed salt (using XPS) experiments. Mixed ion sorption tests
were done with a solution of KCl and CsCl (0.25 M each) at pH 3.2. Data are pre-
sented as mean ± SD from $n = 3$ independent measurements (**C**, **D**).

using mixed ion solutions, quantified by XPS, revealed an even more
pronounced preference for $K^+$ (Fig. 3D). This heightened selectivity
likely arises from the competitive environment created when both $K^+$
and $Cs^+$ ions are present, with $K^+$ demonstrating a stronger affinity for
the crown ether binding sites. When $K^+$ ions are selectively
complexed by the crown ether moieties, they also effectively repel $Cs^+$
ions from entering the polymeric structure and interacting with sites
that would have otherwise been accessible in single-ion sorption
experiments.

**Selective ion transport in crown ether-based polyamide films**
Selective ion transport measurements showed that the crown ether-
based polyamide films preferentially transported $K^+$ over other
monovalent and divalent cations (Fig. 4). To assess this selectivity, a
series of experiments were conducted in a diffusion cell. In each
experiment, the feed solution contained a binary salt mixture with
equimolar amounts of KCl and the chloride salt of a competing cation
at pH 6, while the receiving side was filled with deionized water
(Fig. 4A). The concentration of each ion in the receiving side was
monitored over time to determine their respective transport rates. The
competing cation in the binary mixture was varied between experi-
ments, allowing us to evaluate $K^+$ selectivity relative to different
monovalent and divalent cations. In these experiments, the crown
ether-based polyamide films were mounted on isotropic anodic alu-
minum oxide (AAO) membranes with 20–30 nm pores for mechanical

support. To enhance adhesion and prevent delamination, the AAO
supports were first coated with a 1.5 bilayer polyelectrolyte gutter layer
prior to film transfer. AAO supports were used as a rigid, well-defined
model substrate to enable transport measurements of the ultrathin
selective layer; they are not intended to represent pressure-bearing
supports for practical membrane operation. SEM surface analysis of
the composite membrane shows the thin crown ether-based poly-
amide film positioned over the polyelectrolyte-coated AAO pores
(Fig. 4B). The thinness of both the crown ether-based polyamide film
and the polyelectrolyte layer allowed the electron beam to penetrate
through them, enabling imaging of the underlying AAO pores.

To assess the support's contribution to ion transport, we first
measured the selectivity and ion flux of the polyelectrolyte-coated
AAO support alone for each cation pair of interest. These baseline
measurements allowed us to compare the performance of the support
with and without the crown ether-based polyamide selective layer
(Supplementary Fig. 7). Upon covering the polyelectrolyte-coated AAO
with the crosslinked crown ether-based polyamide film, we observed a
substantial decrease in ion flux and a marked increase in ion selectivity,
as expected. Specifically, the ion flux dropped by approximately two
orders of magnitude, and $K^+$ selectivity increased across all ion pairs
tested. These results confirm that the crown ether layer governs the
transport of ionic species, with minimal contribution from the support.

To place the reduced ion flux in context, we compared the $K^+$ flux
and selectivity of the present membrane with those of reported $K^+$-
selective membranes (Supplementary Table 5). While the $K^+$ flux of the
crown ether–based polyamide is lower than that of many reported
systems, it falls within a range expected for polymeric membranes that
rely on binding-mediated selectivity rather than continuous ion-
conducting pathways. The reduced flux reflects transport through a
dense, highly crosslinked polyamide network with a non-ordered dis-
tribution of crown ether binding sites, some of which may not be
accessible during ion transport.

Ion transport through the crown ether-based polyamide film can
occur via two primary pathways: (i) through interconnected channels
or free-volume elements within the polyamide matrix and (ii) through
specific, reversible interactions with the crown ether moieties (Fig. 4C).
In an ideal membrane with perfectly arranged crown ether sites and
optimal confinement, the latter pathway would dominate. However, in
the prepared membranes where there is a random arrangement, both
pathways likely contribute. $K^+$ ions can utilize both transport pathways,
while competing ions such as $Li^+$, $Cs^+$, and $Mg^{2+}$ primarily rely on the
free-volume pathway due to their weaker affinity for 18-crown-6,
resulting in lower flux. Furthermore, in multi-salt tests, when $K^+$ ions
selectively bind to the crown ether cavities, they will induce charge
repulsion effects that further limit available transport pathways for
competing cations.

Testing a crown ether-based polyamide membrane with various
cation pairs revealed selectivity for most pairs, except $K^+/Na^+$, for
which it was not selective (Fig. 4D). The inability of the membrane to
differentiate between potassium and sodium ions is not surprising
given the relatively low affinity of dibenzo-18-crown-6 for $K^+$ over $Na^+$
(Fig. 1A). The $K^+/Mg^{2+}$ selectivity was conservatively estimated to be
>300, based on the instrument's detection limit, as no $Mg^{2+}$ was
detected on the receiving side after 3 days of testing. The measured $K^+/$
$Li^+$ and $K^+/Cs^+$ selectivities were 3.8 and 4.4, respectively. Achieving
such high selectivities between similar monovalent ions, where charge-
based mechanisms offer no advantage, is rare in polymeric mem-
branes. The minimal differences in their hydrated radii—only 1.02 Å
between $K^+$ and $Li^+$, and 0.04 Å between $K^+$ and $Cs^+$ (Fig. 1A)—make size-
based differentiation exceptionally challenging. A separation
mechanism involving partial dehydration, where water molecules are
removed to reduce ion size, could favor either $Li^+$ or $Cs^+$, depending on
the average pore size, as $K^+$ occupies an intermediate position in both
bare ion size and dehydration energy. The precise separations we

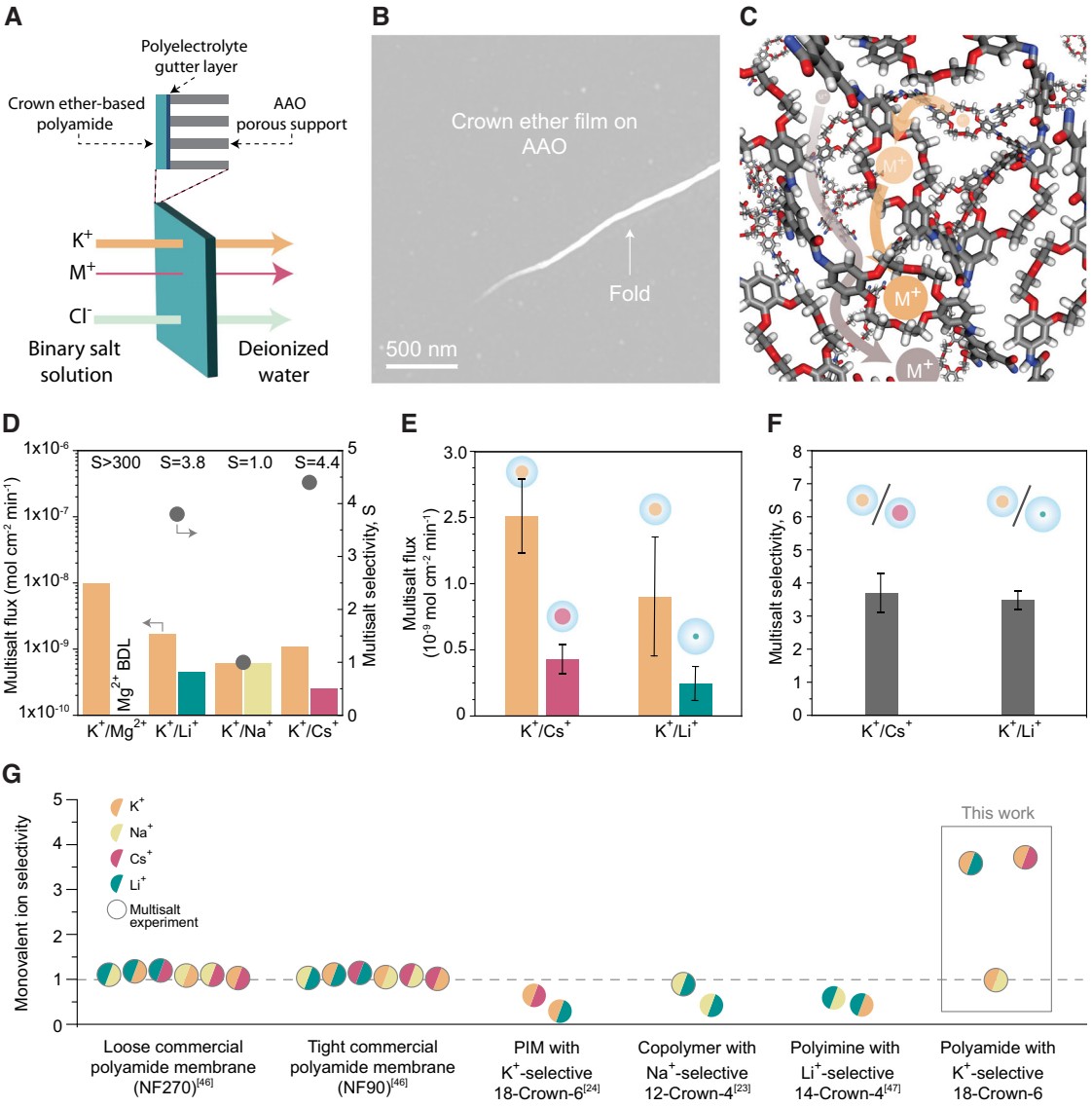

**Fig. 4 | Selective monovalent ion transport through the crown ether-based polyamide film. A** Schematic of the composite membrane's structure and the measurement method for evaluating salt transport across crown ether-based polyamide films. AAO stands for anodic aluminum oxide, and the polyelectrolyte gutter layer was made via layer-by-layer deposition of polyacrylic acid/ poly-ethyleneimine/polyacrylic acid. The composite membrane was mounted between the feed and receiving chambers, which were temperature-controlled via a water jacket. All experiments were done with binary cation mixtures prepared with chloride salts in the feed, deionized (DI) water as the receiving solution, and a temperature of 25 °C; **B** SEM image of the composite membrane. The AAO pores are visible thanks to the thinness of the crown ether-based polyamide; **C** schematic highlighting two potential ion transport pathways: (i) through interconnected free-volume elements (i.e., pores) available in the crosslinked structure, depicted in gray, and (ii) through reversible interactions with crown ether moieties, depicted in orange; **D** cation flux (left) and K$^+$ selectivity over a competing cation (right) for a

crown ether-based polyamide composite membrane submitted to consecutive multi-salt tests. BDL stands for below the detection limit of the ion chromatography system. The minimum K$^+$/Mg$^{2+}$ selectivity was conservatively calculated by using the detection limit. **E** Average flux of monovalent cations across three membranes under two multi-salt feed conditions: 0.1 M KCl combined with 0.1 M CsCl or LiCl. **F** Average selectivity of K$^+$ over other monovalent cations (Cs$^+$ or Li$^+$) in these multi-salt feed solutions. **G** Monovalent cation selectivities for polymeric membranes using crown ether molecules as their main building block. Loose (FilmTec™ NF270 from DuPont) and tight (FilmTec™ NF90 from DuPont) commercial polyamide membranes lacking crown ether moieties are included for comparison. Selectivity is quantified by the flux ratio of the crown ether's preferred cation to that of a competing cation. Data derived from multi-salt experiments are marked by a gray contour, while data from single-salt experiments do not have a contour. Referenced data sources are: refs. 23,24,30,46. Data are presented as mean ± SD from $n \geq 3$ independent measurements (**E**, **F**).

observe, favoring K$^+$, likely result from molecular recognition through coordination chemistry, enabled by a facilitated transport mechanism.

The K$^+$ ion flux was significantly higher when Mg$^{2+}$ was the competing cation ($1 \times 10^{-8}$ mol cm$^{-2}$ min$^{-1}$) than when the competing cations were monovalent (1.7, 0.6, 1.1 ×10$^{-9}$ mol cm$^{-2}$ min$^{-1}$ for Li$^+$, Na$^+$, and Cs$^+$, respectively). This increase results from the higher Cl$^-$ concentration in the feed and the membrane's high K$^+$/Mg$^{2+}$selectivity. The elevated Cl$^-$ flux generates a membrane potential that drives the

transport of cations—primarily K$^+$ due to the high selectivity—across the membrane to maintain charge neutrality. Consistent with this, the lowest K$^+$ flux was observed for the least selective ion pair (K$^+$/Na$^+$), where both cations could readily transport through the membrane to maintain electroneutrality.

Preparing polyamide films at a free interface and then transferring them to a support, as done in this study, enables characterization using multiple techniques (AFM, TEM, XPS, GIWAXS, QCM) on the same

selective layer used for transport studies. However, a key drawback of this method is that defects introduced during transfer can significantly impact transport studies, even though they do not affect other characterization techniques. For example, a crack in the film during transfer or debris on the porous support that stretches the film upon drying can create pinholes, resulting in non-selective transport pathways that dominate ion transport (Supplementary Fig. 8). To obtain statistically significant data on the transport properties of the prepared films, we fabricated enough membranes to secure at least three high-quality samples for each of the two key transport tests: mixed salt diffusion experiments for $K^+/Cs^+$ and $K^+/Li^+$. These tests were specifically selected to confirm that the crown ether building blocks impart selective transport properties to the membrane, favoring ions with which they form stronger complexes (i.e., $K^+$). High-quality films were identified as those showing a $K^+$ flux more than an order of magnitude lower than that of the bare support during $K^+/Cs^+$ and $K^+/Li^+$ experiments (Supplementary Fig. 7). The full transport data for all measured membranes, including those not meeting the integrity criterion, are provided in the Supporting Information (Supplementary Tables 1–3). Results indicated that $K^+$ flux ranged from 0.6 to $2 \times 10^{-8}$ mol cm$^{-2}$ min$^{-1}$ in these membranes (Fig. 4E), with average mixed salt selectivities of 3.5 for $K^+/Li^+$ and 3.7 for $K^+/Cs^+$ (Fig. 4F).

When benchmarked against commercial polyamide membranes made by IP (e.g., FilmTec™ NF270 and NF90), the ultrathin crown ether-based polyimide films prepared here demonstrated higher monovalent ion selectivity, attributed to the crown ether moieties (Fig. 4G). Previous efforts to use crown ether molecules as building blocks for ion-selective polymeric membranes have leveraged complexation at crown ether sites as "sinks" where ions forming the most energetically favorable complexes are delayed, often resulting in inverse selectivities towards the ion that formed the most favorable complex with the crown ether building block (Fig. 4G). This effect is likely due to the large thickness of these membranes (on the order of tens of micrometers), which allows for numerous reversible ion–crown ether interactions before an ion can cross to the opposite side. The "sink" effect was further intensified, in some cases, by a low density of crown ether moieties, which hindered efficient ion hopping. In contrast, the crown ether-based polyamide films prepared here are only 6 nm thick, meaning that an ion traveling a direct path would interact with fewer than 18 times with crown ether sites—18 being the theoretical maximum if all crown ether moieties were perfectly aligned, which is not the case in our system. This minimal path length, combined with the high density of crown ether sites, enabled the films to selectively partition and efficiently transport the ion that forms the most favorable complex with the crown ether building blocks, achieving greater transport efficiency than competing monovalent and divalent ions. As a result, these polymeric membranes exhibited monovalent ion-ion selectivity toward the ion with the strongest crown ether complexation (Fig. 4G).

To assess the behavior of the crown ether–based selective layer under pressure-driven conditions, membranes were also fabricated directly on porous polyacrylonitrile polymeric supports using identical IP conditions. Water permeance and salt rejection measurements were performed in a stirred cell at 3.5 bar, and high rejection of Rose Bengal (>95%) was observed, confirming complete surface coverage and the absence of macroscopic defects in the selective layer. As summarized in Supplementary Table 4, these membranes exhibit water permeances of approximately 1.3 L m$^{-2}$ h$^{-1}$ bar$^{-1}$, and a moderate rejection of Mg$^{2+}$ (~32%), while showing low rejection (8–10%) for monovalent $K^+$ and $Li^+$ ions under both single-salt and mixed-salt conditions. The pressure-driven transport behavior observed here is consistent with that of conventional nanofiltration-type polyamide membranes, including those reported by Alhazmi et al., who employed the same amino-functionalized crown ether monomer in IP on polymeric supports and optimized fabrication conditions to leverage the intrinsic macrocyclic

cavity for pharmaceutical separations[31]. These results indicate that under pressure-driven operation, convective solvent transport can dominate ion transport, limiting discrimination among ions of similar size and charge.

## Discussion

This study demonstrates that crown ether-based polymeric membranes can achieve selective ion transport favoring the ion preferred by the crown ether, provided they are fabricated as ultrathin films with a high density of ion-binding sites. Using IP to incorporate 18-crown-6 molecules, we produced membranes made entirely from crosslinked crown ether building blocks, achieving preferential $K^+$ transport over other cations with ion-ion selectivities of approximately 4 for monovalent cation pairs ($K^+/Cs^+$ and $K^+/Li^+$). Sorption experiments revealed a distinct preference for $K^+$, particularly in mixed-salt environments where competitive effects amplified the selectivity of the crown ether moieties. By challenging the membranes to selectively transport $K^+$ against competing monovalent cations with smaller ionic and hydrated sizes and lower dehydration energies, we consistently observed $K^+$ preferential transport, confirming that reversible complexation with crown ether sites was the primary mechanism driving this selectivity. These findings underscore the critical role of selective ion complexation in governing both sorption and transport performance in these membranes.

While the monovalent ion selectivity achieved by the prepared membranes is impressive for a polymeric system, it falls short of the higher selectivities observed in highly ordered crystalline channels such as crown ether-functionalized MOFs[22], where the spatial arrangement of crown ether molecules is precisely controlled. However, translating these crystalline channels into continuous membranes remains challenging, as ion transport in polycrystalline systems is typically dominated by grain boundaries and defects rather than by the intrinsic selectivity of the channels. In contrast, polymeric membranes are readily scalable, which motivates efforts to enhance their selectivity and transport rates to approach those of crystalline systems. Bridging the performance gap between highly ordered but difficult-to-scale crystalline systems and polymeric membranes fabricated via scalable IP could be achieved by aligning crown ether molecules during polymerization. Recent advances with cyclodextrin-based building blocks have demonstrated that functionalizing them to promote their alignment at the water/organic interface during IP can produce crosslinked films of aligned macrocycle molecules with significantly improved solute–solute selectivities compared to disordered films[33]. Applying similar alignment strategies to crown ether building blocks could enhance the molecular ordering in the resulting membranes, narrowing the selectivity gap while preserving the scalability and processability that make polymeric systems so attractive. In this context, more stringent control over free volume distribution and crown ether alignment, such that continuous water pathways allowing ions to be transported together with water molecules are suppressed, may enable binding-mediated ion selectivity to persist under pressure-driven conditions.

In addition, the use of different crown ether regioisomers offers an unexplored handle to tune network topology, local packing, and ion–ligand accessibility in polymeric membranes. Likewise, varying the structure of the acid chloride comonomer, such as its rigidity or length, could provide a means to systematically control the spacing and packing density of crown ether units within the polyamide network. Together, these approaches offer complementary strategies to engineer crown ether–based membranes with higher degrees of molecular order and, consequently, enhanced ion selectivity and ion transport, while preserving the practical advantages of IP.

This study lays the groundwork for leveraging the intrinsic ion-selective properties of crown ethers to develop scalable polymeric membranes for ion separation. Future efforts should focus on

expanding this approach to incorporate other macrocyclic building blocks, such as crown ethers selective for critical materials like $Li^+$ or macrocycles specifically designed for complexing rare earth elements. Additionally, controlling the spatial arrangement of these building blocks during polymerization to create more ordered structures and testing the membranes under conditions that mimic industrial applications will be crucial to fully demonstrate their potential.

## Methods

### Materials

AAO membranes with a diameter of 25 mm and an average pore size of less than 20 nm were obtained from Hefei Pu-Yuan Nanotechnology Ltd. Dibenzo-18-crown-6, hydrazine monohydrate, palladium on carbon (10 wt.% Pd/C), and N,N-dimethylformamide (DMF) were supplied by Sigma-Aldrich. 1,3,5-Benzenetricarbonyl chloride (TMC) was acquired from Thermo Scientific Chemicals. Lithium chloride (LiCl) was sourced from Acros Organics; sodium chloride (NaCl) and potassium chloride (KCl) were obtained from Sigma-Aldrich; cesium chloride (CsCl) and magnesium chloride ($MgCl_2$) were purchased from Thermo Scientific Chemicals. Poly(ethyleneimine) (PEI) solution (50 wt.% in water, average molecular weight ~750,000 g/mol) and polyacrylic acid (PAA) solution (35 wt.% in water, average molecular weight ~100,000 g/mol) were purchased from Sigma-Aldrich. All aqueous solutions were prepared using ultrapure water from a Milli-Q purification system.

### Synthesis of trans-di(aminobenzo)-18-crown-6 (DAB18C6)

The synthesis of DAB18C6 was performed following the methodology reported by Patel et al., starting from commercially available dibenzo-18-crown-6[35]. The process involved two main steps: the nitration of dibenzo-18-crown-6 to obtain trans-di(nitrobenzo)-18-crown-6, followed by the reduction of the nitro groups to yield the target compound, DAB18C6. All reactions were carried out under a nitrogen atmosphere unless otherwise specified, and the progress of the reactions was monitored by thin-layer chromatography where applicable. NMR spectra were recorded in $CDCl_3$, and chemical shifts ($\delta$) are reported in parts per million (ppm). This synthesis route effectively converts dibenzo-18-crown-6 into an amino-functionalized crown ether, providing the key building block for our studies.

**Step 1.** Synthesis of trans-di(nitrobenzo)-18-crown-6. Dibenzo-18-crown-6 (10 g, 27.8 mmol) was dissolved in dichloromethane (DCM, 150 mL). Acetic acid (120 mL) was added sequentially to the solution. Concentrated nitric acid (69%, 15 mL) was then added dropwise under continuous stirring at room temperature. The reaction mixture was stirred for 48 h at room temperature, during which nitration of the aromatic rings produced a mixture of cis- and trans-di(nitrobenzo)-18-crown-6 isomers. Owing to its significantly lower solubility in the reaction medium, the trans isomer selectively precipitated from solution[35]. The precipitated product was collected by filtration and washed sequentially with DCM (100 mL) and deionized water (50 mL). The solid was dried under vacuum at 70 °C for 24 h to afford trans-di(nitrobenzo)-18-crown-6 as a white powder (9.6 g, 77% yield, $^1H/^{13}C$ NMR spectrum available in Supplementary Fig. 1A, B). $^1H$ NMR (400 MHz, $CDCl_3$, $\delta$): 7.85 (d, 2H; $J = 9$ Hz), 7.68 (s, 2H), 7.12 (d, 2H; $J = 9$ Hz), 4.18 (br, 8H), 3.82 (br, 8H). $^{13}C$ NMR (100 MHz, $CDCl_3$, $\delta$): 154.2, 148.1, 141.0, 118.0, 111.7, 107.0, 68.8-6.84. The $^1H$ and $^{13}C$ NMR data are consistent with previously reported characterization for this compound[35].

**Step 2.** Synthesis of DAB18C6. Trans-di(nitrobenzo)-18-crown-6 (5 g, 11.1 mmol) and 10% palladium on carbon (Pd/C, 450 mg) were added to ethanol (150 mL) under a nitrogen atmosphere. Hydrazine monohydrate ($NH_2NH_2·H_2O$, 30 mL) was then added to the mixture. The reaction mixture was heated to reflux at 90 °C and stirred for 5 h. Upon

completion, the hot mixture was immediately filtered to remove the catalyst and prevent crystallization of the product. The filtrate was concentrated using a rotary evaporator to remove the solvent. The resulting solid was dried under vacuum at 80 °C for 24 h to obtain DAB18C6 as a powder (3.8 g, 88% yield, $^1H/^{13}C$ NMR spectrum available in Supplementary Fig. 1C, D). $^1H$ NMR (400 MHz, $CDCl_3$, $\delta$): 6.59 (d, 2H; $J = 8$ Hz), 6.20 (s, 2H), 6.02 (d, 2H; $J = 8$ Hz), 4.63 (s, 4H), 3.90 (m, 8H), 3.75 (m, 8H). $^{13}C$ NMR (100 MHz, $CDCl_3$, $\delta$): 149.5, 143.8, 139.6, 116.0, 105.8, 101.3, 69.6-68.2. The $^1H$ and $^{13}C$ NMR data are consistent with previously reported characterization for this compound[35].

### Synthesis of crown ether-based polyamide membranes

**Preparation of solutions.** DAB18C6 was initially dissolved in dimethylformamide (DMF) to prepare a 1% (w/v) stock solution. This stock solution was then diluted with Milli-Q water to achieve a final concentration of 0.2% (w/v), using a solvent mixture of 20% DMF and 80% water (v/v). Similarly, trimesoyl chloride (TMC) was dissolved in hexane to prepare a 0.2% (w/v) solution and stirred for 30–60 min to ensure complete dissolution.

**Film formation.** A silicon wafer with a rod attached perpendicularly at one corner was placed inside a Petri dish. Approximately 7–10 mL of the DAB18C6 solution was poured into the dish to fully cover the silicon wafer, ensuring a solution depth of at least ~1 cm. Three to five milliliters of the TMC/hexane solution were then carefully poured from the side of the Petri dish to minimize disturbance of the interface during the pouring process. After a defined reaction time, the crown ether-based polyamide membrane formed at the interface.

**Film collection and drying.** The interfacial membrane was carefully retrieved using the silicon wafer by manipulating it via the attached rod. The wafer was approached to the interface with a continuous movement while maintaining a slight angle to facilitate the lifting of the membrane. Once the film adhered to the wafer and was taken out of the Petri dish, it was transferred to a large Milli-Q water bath to float freely. After 5–10 min, the floating thin films were collected onto different substrates: porous substrates for transport studies, gold-coated glass slides for XPS, silicon wafers for AFM, and Quantifoil R 1.2/1.3 Holey Carbon grids for TEM. The supported films were dried overnight at room temperature to ensure good adhesion to the substrates.

**Post-treatment.** For XPS and AFM analyses, the samples were further rinsed sequentially with hexane, water, and isopropanol, followed by an additional drying step. The samples intended for transport studies were used after the initial drying step, within 1 week of preparation.

**Porous supports used for transport studies.** To provide additional mechanical reinforcement to the ultrathin crown ether-based polyamide membrane, we utilized AAO membranes with small pores. To improve the interaction between the membrane and the AAO supports, we coated the AAO with a "cushion" polyelectrolyte layer. We observed that directly depositing the film onto AAO supports without the polyelectrolyte layer resulted in poor adhesion; the film would often delaminate from some sections when immersed in water after drying. The polyelectrolyte "cushion" has high permeance to ions, allowing us to effectively probe the transport properties of the crown ether-based polyamide membrane. To coat the AAO with the polyelectrolyte layer, the AAO membrane surface underwent 1.5 cycles of successive treatments with polyelectrolytes—specifically poly(acrylic acid) (PAA) and poly(ethyleneimine) (PEI)—following the surface coating procedures described in the literature[43,44]. Each full cycle consisted of sequential deposition of 0.1 wt% PAA (pH ~3.5) for 10 min and 0.1 wt% PEI (pH ~10.6) for 15 min onto the surface. After each polyelectrolyte deposition, the membrane was immersed in ultrapure

water for 5 min to remove any unbound molecules. The half-cycle involved a final deposition of PAA only.

**Notes.** The ultrathin films are delicate, and the manual handling process decreases the success rate. Cracks in the films were often observed during manipulation, leading us to discard these samples immediately. In some cases, however, cracks were only detected after conducting transport studies, as evidenced by an ion flux one to two orders of magnitude higher than when a continuous film covered the pores. As expected, no ion selectivity was observed for films with cracks. An example of the defects present in these films is shown in Supplementary Fig. 8.

### Ion adsorption studies

The freestanding crown ether-based polyamide membranes were transferred onto 5 MHz gold QCM sensors (QSX 301, Biolin Scientific) for single ion sorption studies using QCM, and onto gold-coated glass for mixed ion sorption studies using XPS.

**Single ion.** The coated QCM sensors were first equilibrated in ultrapure water before being sequentially exposed to 0.5 M CsCl and 0.5 M KCl solutions, with ultrapure water washes in between (all solutions at pH 3.2). The frequency changes upon exposure to the ion-containing solutions were monitored using a Q-sense flow module with four parallel channels (Biolin Scientific). Each experiment involved two coated sensors and two bare sensors as blanks, running in parallel. To account for viscosity and density variations in the solutions, the frequency shifts observed with the bare sensors were subtracted from those of the coated sensors. The corrected frequency changes were then used to calculate the mass adsorbed by the crown ether-based polyamide membranes in response to CsCl and KCl solutions, applying the Sauerbrey equation:

$$\Delta f = -\frac{2f_0^2}{A\sqrt{\rho_q \mu_q}} \times \Delta m \tag{1}$$

where $\Delta m$ is the change in mass adsorbed (kg) on the sensor, $\Delta f$ is the change in frequency (Hz), $f_0$ is the resonant frequency (Hz) of the sensor, $A$ is the piezoelectrically active area (m$^2$), $\rho_q$ is the density of quartz (kg m$^{-3}$), and $\mu_q$ is the shear modulus of the quartz (Pa).

**Mixed ion.** Gold substrates coated with crown ether-based polyamide membranes were immersed in a mixed ion solution of KCl and CsCl (0.25 M each, pH 3.2) for 2 days. After removal, they were briefly rinsed in ultrapure water to remove excess solution and left to dry at room temperature overnight. XPS analysis (VersaProbe II, Physical Electronics) was then conducted to quantify the amounts of Cs and K absorbed by the crown ether-based polyamide membranes.

### Ion transport studies

**Diffusion cell setup.** A custom-made diffusion cell with two 60 mL chambers and a water jacket was used for membrane performance testing. The membrane was initially positioned between two stainless steel washers, each featuring a circular opening of 0.4 cm$^2$. To ensure a proper seal, a rubber gasket was placed on both sides of the membrane. The entire assembly was then secured using binder clips, which facilitated easier handling. It was then positioned between the two chambers with its active layer facing the feed solution. The exposed area of the membrane (active area) was 0.4 cm$^2$. The water jacket was maintained at 25 °C using a cooling/heating recirculator (Cole-Parmer Polystat) to control the internal solution temperature. Prior to testing, approximately 20 min were allowed for the recirculator to stabilize the diffusion cell temperature.

**Testing procedure.** All tests done were multi-salt tests. The feed solution contained 0.1 M of two salts: KCl and either NaCl, CsCl, LiCl, or MgCl$_2$, while the receiving solution contained deionized water. Each chamber was continuously stirred with a magnetic stir bar to ensure uniform concentration throughout the solution. At specific time intervals, 5 mL aliquots were withdrawn from the receiving chamber to measure ion concentrations. The samples were analyzed using ion chromatography (IC) (940 Professional IC Vario, Metrohm). The peak areas obtained from the IC signals were converted to salt concentrations by referencing calibration curves established for each specific ion pair. Between tests with different ion pairs on the same membrane, a thorough rinse was performed by flushing both sides of the diffusion cell with deionized water three times. The final rinse involved stirring in deionized water for over an hour.

The ion flux ($J_c$) was calculated using Eq. (2):

$$J_c = \frac{C_c V}{A t} \tag{2}$$

where: $C_c$ is the ion concentration in the receiving solution, $V$ is the volume of the receiving solution, $A$ is the active membrane area during testing, and $t$ is the duration of the experiment.

Ion selectivity was subsequently determined using Eq. (3):

$$\text{Selectivity} = \frac{J_i}{J_j} \cdot \frac{C_j}{C_i} \tag{3}$$

where: $J_i$ and $J_j$ are the fluxes of the target ion and the competing ion, respectively, and $C_i$ and $C_j$ are the concentrations of the target ion and the competing ion in the feed solution. In this study, potassium ion (K$^+$) was selected as the target ion due to its highest binding affinity to 18-crown-6, while the other cations (i.e., Cs$^+$, Li$^+$, Mg$^{2+}$) served as competing ions.

### Characterizations

**FTIR.** To prepare the crosslinked crown ether-based polyamide samples for FTIR, 10 mL of a 0.2% (w/v) TMC solution in hexane was contacted with 10 mL of a 0.2% (w/v) DAB18C6 solution in DMF-water (20/80 v/v) and shaken for 1 h to continuously generate a new interface. The crosslinked polyamide that formed at the interface was then recovered via filtration and thoroughly washed with hexane and isopropanol. DAB18C6 powder was used as is. Both samples were dried at 60°C for 24 h before FTIR measurements were performed using a Shimadzu IRTracer-10 with 20 scans at a resolution of 2 cm$^{-1}$.

**XPS.** To prepare the samples for XPS, crown ether-based polyamide membranes were formed at a free interface, as described in the methods section, with a reaction time of 5 min, and then transferred onto a gold-coated glass slide. The membranes were thoroughly washed with hexane, water, and isopropanol, followed by drying for at least 24 h. XPS analysis was performed using a VersaProbe II (Physical Electronics). Elemental signals were collected with a 200-μm X-ray beam, 50 W power, and 15 kV electron beam energy. A survey scan was conducted five times for each membrane to identify elements, using a pass energy of 187.85 eV and a step size of 0.8 eV. High-resolution scans were taken for carbon, oxygen, nitrogen, and, for ion adsorption studies, potassium and cesium, with each scan repeated 5, 10, 20, and 20 times, respectively. The pass energy and step size for these scans were set to 23.5 eV and 0.1 eV. All data were processed using CasaXPS (Version 2.3.25PR1.0), with calibration based on the C 1s peak at 284.4 eV. Bond deconvolution for oxygen, nitrogen, and silicon followed literature guidelines[45].

**SEM.** A field emission scanning electron microscope (FE-SEM, Hitachi SU-70, Hitachi High Technologies America) was used for sample

imaging. Before imaging, all samples were coated with an 8 nm iridium layer using a Cressington 208 sputter coater (Ted Pella). SEM images were acquired at 5 kV with a working distance of 10–15 mm.

**TEM.** Cryo-S/TEM imaging was performed using a JEOL JEM-2100F transmission electron microscope (TEM). Cryo-TEM images were formed on a Gatan ORIUS SC1000 charge-coupled device camera. Cryogenic low-angle annular dark-field scanning transmission electron microscopy (LAADF-STEM) images were collected using a Gatan Model 805 detector with a 12–20 cm camera length. Cryogenic HAADF-STEM and BF-STEM images were collected using JEOL detectors (EM-24560 and EM-24541SIOD, respectively) with a camera length of 3–10 cm. For all STEM imaging, dwell times between 20 and 25 μs were used with the preset 1 nm spot size. All imaging was performed at cryogenic temperatures using a Simple Origin Model 215 cryo transfer holder. In all cases, the holder was loaded into the TEM column and then cooled to cryogenic temperature. TEM images were collected at magnifications below 40,000×. STEM images were collected at magnifications of 60,000–600,000× with a scan size of 1024 × 1024 pixels.

**AFM.** To prepare the samples for AFM, crown ether-based polyamide membranes were formed at a free interface, as detailed in the methods section, with varying reaction times, and then transferred onto Si-wafers. The membranes were thoroughly washed with hexane, water, and isopropanol, dried for at least 24 h, and then scratched with a needle. AFM measurements were performed using a Bruker Dimension FastScan with a FastScan-B tip (5 nm tip radius, Bruker, Billerica, MA) in tapping mode. For each condition, at least eight height measurements were recorded. Data processing was carried out using Gwyddion software. The images were flattened prior to height measurements.

**GIWAXS.** The crown ether-based polyamide film was transferred to a Si wafer, washed with hexane, water, and isopropanol, then dried overnight prior to measurement. Measurements were conducted using a Xenocs Xeuss 3.0 equipped with a microfocus sealed-tube Cu 30 W/30 μm X-ray source (Cu K-α, $\lambda = 1.54$ Å).

## Data availability
All data that support the findings of this study are available within the paper and its Supplementary Information or from the corresponding authors upon request.

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

## Acknowledgements

This work was supported as part of the Center for Enhanced Nanofluidic Transport (CENT), an Energy Frontier Research Center funded by the U.S. Department of Energy, Office of Science, Basic Energy Sciences under Award # No. DE-SC0019112 (M.E.). L.F.V. acknowledges support from the Ershaghi Center for Energy Transition Seed Fund Program and a post-doctoral fellowship from the Swiss National Science Foundation (P400P2_199330). The authors thank J. Karosas for assistance with ICP-MS at the Yale Analytical and Stable Isotope Center (YASIC). Yale Institute of Nanoscale and Quantum Engineering (YINQE) provided AFM and SEM equipment. Yale Chemical and Biophysical Instrumentation Center (CBIC) provided FTIR equipment. Yale Glass Shop provided diffusion cells.

## Author contributions

L.F.V. and M.E. conceptualized and designed the study. L.F.V. synthesized and characterized the membranes and performed SEM and QCM characterizations. J.Z. performed the XPS analysis. J.L. synthesized the crown ether-based monomer under the guidance of M.Z. A.T.H. performed TEM under the guidance of J.C. R.M.D. and C.V. contributed to data analysis and interpretation. L.F.V. and M.E. wrote the manuscript, with all authors contributing to manuscript editing.

## Competing interests

The authors declare no competing interests.
