## [Transparent Peer Review file · Nature Communications]

Ultrathin crown ether-based polyamide membrane for ion-ion separations

Corresponding Author: Professor Luis Francisco Villalobos

Version 0:

Reviewer comments:

Reviewer #1

(Remarks to the Author)

This manuscript focuses on the fabrication of crown ether polyamide membranes for ion separation. Crown ethers have been of research interest for molecular separation due to the intrinsic pore structures. The authors demonstrated ultrathin membranes prepared via interfacial polymerisation, a common method for commercial polyamide membranes. Their novel investigation and characterisation techniques achieve high separation performance for K⁺ ions, though the experimental scope is limited. Further investigations are recommended as per the comments below to validate the findings.

- There're different types of crown ethers with the different number of atoms in the ring. The selection criteria for dibenzo-18-crown-6-ether should be clarified, considering that while 18-crown-6-ether binds K⁺ ions, the non-planar structure of polyamide films and polyamide bonds may affect the pore size.

- Water contact angle data is critical for understanding membrane permeation behaviour in aqueous ion separation applications.

- The nitration of dibenzo-18-crown-6 should yield both cis- and trans-DAB18C6. The methodology for their separation and the rationale for using only trans-DAB18C6 in this study should be detailed.

- Have you tried the same polyamide membranes from cis- DAB18C6? How were the results?

- The polyelectrolyte gutter layer is essential in this study for preventing membrane cracks and leakage. Additional characterisation, including thickness and water and ion permeability, is necessary. Why was a 1.5-layer coating of PEI and PAA selected?

- The crown-ether polyamide layer demonstrated stability on PVDF porous support (Fig S2d). Was a gutter layer also applied to this sample?

- The mechanical properties of the selective layer are critical for membrane separation applications, despite minimised hydraulic pressure on ion separations. Can the membrane withstand pressure on an AAO substrate?

- How does membrane performance vary with thickness? A thicker layer might decrease permeance but enhance selectivity.

- Ion permeability performance along with selectivity should be compared with existing literature data.

- The current fabrication method is complex. Given that interfacial polymerisation (IP) for RO membranes is directly applied to substrates for simplicity, can this crown ether polyamide layer also be directly applied? If so, how does it affect separation performance.

Reviewer #2

(Remarks to the Author)

In this manuscript, the authors developed an ultra-thin, crown ether functionalized membrane using interfacial polymerization. Due to the specific interaction between 18-crown-6 and K⁺ ions, along with the ultra-low membrane thickness of 6.6 nm, the membrane shows good selectivity between monovalent cations of 3.8 for K⁺/Li⁺ and 4.4 for K⁺/Cs⁺. The proposed mechanism involves reversible complexation between the crown ether sites and K⁺ ions. Overall, the work presents an interesting concept that could appeal to the membrane community. However, there are still some major issues that need to be addressed before considering its publication in Nature Communications.

1. Figure 2h shows a rather low signal-to-noise ratio, which makes it difficult to clearly support the presence of unreacted –NH₂ groups. Additionally, the authors should consider discussing the potential impact of hydrolyzed –COOH or protonated –NH₃⁺ groups (under low pH conditions) on ion flux and selectivity, as these groups could facilitate the transport of anions or

cations.

2. In Figure 3d, the author used XPS to quantify the adsorption of different ions. However, since XPS only probes the top few tens of nanometers, it mainly reflects surface information. It's strongly recommended to use ICP or other bulk characterization methods to support this analysis.

3. The transfer process after interfacial polymerization introduces defects and hinders scalability. Could direct polymerization on a porous membrane (polymeric or ceramic) achieve comparable selectivity? In addition, on Page 9, line 22, the author mentions testing three "high-quality" samples for selectivity. How were these samples selected? Were they the top-performing membranes, or was there another criterion?

4. The author mentions the membrane's mechanical robustness. Could this claim be supported up by experimental data, like AFM measurements?

5. The ion absorption tests were done at low pH, were the selectivity measurements performed under the same acidic conditions? Also, would this low pH cause amide group hydrolysis and decrease long-term stability?

6. Even though the membrane is very thin, the flux dropped by 1-2 orders of magnitude compared to the bare AAO substrate, this might be because it lacks ion-conducting groups. How does this flux performance compare to other ion-selective membranes? Also, could the authors suggest possible ways to improve the flux without compromising selectivity?

7. Given the extensive literature on the use of crown ethers in interfacial polymerization for ion-selective membranes [Angewandte Chemie, 2023, 62(18): e202300167; Journal of Materials Chemistry A, 2025, 13(16): 11732-11748; Journal of Membrane Science, 2025, 714: 123372], the novelty and uniqueness of this study should be further clarified and emphasized.

Reviewer #3

(Remarks to the Author)

The paper describes the preparation and characterization of interfacial polymerized thin-film composite membranes with an amino-functionalized crown ether as hydrophilic monomer. The key finding of the work is the preferential K^+ transport over other monovalent cations. The ion-ion selectivities of approximately 4 for monovalent cation pairs (K^+/Cs^+ and K^+/Li^+) is impressive. The conclusions are supported by the well-described experiments.

The paper would have more relevance for applications if the IP films would have been prepared on a polymeric support membrane. Here the support was an anodic aluminium oxide membrane. These membranes are brittle and prevent applications under any mechanical stress like pressure filtration. In addition they are not compatible with the polymeric IP film. This required a polyelectrolyte inter layer which influences the cation transport. The ions transport properties were not measured under filtration condition but in a diffusion cell, which has little relevance for practical applications. The authors should have cited the work of Alhazmi et al. (ref. 32) in a correct way. The membrane preparation part in this reference is very similar, also using interfacial polymerisation and amino-functionalized crown ether as hydrophilic monomer. But they use a polymeric support membrane and pressure filtration for transport characterisation. This should have been discussed in this work.

The authors describe on page 5 that thin membranes favor the high selectivity because the rate limiting step for the transport of the non-favored ion is the partitioning into the membrane, whereas the rate limiting step for the target ion is its movement through the network of binding sites. For me it is not convincing that the solution of the not complexed ion into the water-swollen membrane is rate limiting.

Version 1:

Reviewer comments:

Reviewer #1

(Remarks to the Author)

All comments the reviewer addressed were carefully discussed in the revised manuscript. No further comments and objections to publishing this work.

Reviewer #2

(Remarks to the Author)

The authors have addressed my comments and concerns. The revised version is recommended for publication.

Reviewer #3

(Remarks to the Author)

The authors have addressed my concerns in detail. In particular, it has been clarified that the high selectivity for monovalent cations could not be observed under pressure-driven conditions. While this does reduce the practical relevance of the work, the impressive selectivity under diffusion-controlled conditions nonetheless clearly justifies publication. My critical comment regarding the choice of the support membrane is now also discussed appropriately. In my view, the manuscript can be published without further revisions.

Klaus-V. Peinemann

Response to Reviewers

Color coding used in this document:

- Reviewer comments are shown in **black**.
- **Blue** text indicates the authors' responses to reviewer comments.
- **Orange** text highlights text that was added or modified in the revised manuscript in response to the reviewers' suggestions.

Reviewer #1 (Remarks to the Author):

This manuscript focuses on the fabrication of crown ether polyamide membranes for ion separation. Crown ethers have been of research interest for molecular separation due to the intrinsic pore structures. The authors demonstrated ultrathin membranes prepared via interfacial polymerisation, a common method for commercial polyamide membranes. Their novel investigation and characterisation techniques achieve high separation performance for K^+ ions, though the experimental scope is limited. Further investigations are recommended as per the comments below to validate the findings.

R: We thank the reviewer for their careful assessment and constructive comments. We appreciate their recognition of the novelty of our approach and the separation performance achieved for K^+ ions, and we have taken their suggestions into account by addressing the recommended points to further strengthen and validate the study.

1. There're different types of crown ethers with the different number of atoms in the ring. The selection criteria for dibenzo-18-crown-6-ether should be clarified, considering that while 18-crown-6-ether binds K^+ ions, the non-planar structure of polyamide films and polyamide bonds may affect the pore size.

R: We thank the reviewer for this insightful comment. Dibenzo-18-crown-6 was selected as a model building block to test a fundamental hypothesis: whether a disordered polymeric assembly of crown ether motifs embedded within a cross-linked polyamide network can collectively give rise to ion-selective transport. Our objective was not to optimize K^+ separation specifically but rather to assess, at a fundamental level, whether this design strategy is viable and merits broader consideration for polymeric ion-selective membranes.

Dibenzo-18-crown-6 was chosen because its size and coordination geometry are well matched to K^+ in solution, it is chemically robust, and its diamine-functionalized derivatives are well established, enabling straightforward incorporation via interfacial polymerization without introducing additional synthetic variables. This allowed us to focus on structure–transport relationships rather than monomer development.

We agree with the reviewer that incorporation into a non-planar, densely cross-linked polyamide network can distort the ideal crown-ether geometry and effectively rigidify the macrocycle. As a result, the binding environment experienced by ions in the membrane differs from that of isolated crown ethers in solution. Consistent with this, our adsorption measurements show that dibenzo-18-crown-6 retains a preference for K^+ after incorporation, although the selectivity is reduced relative to the free macrocycle. This behavior is consistent with prior supramolecular studies showing that immobilization and increased rigidity reduce conformational adaptability while preserving size-based recognition^[1].

Accordingly, ion transport in these membranes should not be interpreted as occurring through discrete, idealized crown-ether pores, but rather through a three-dimensional polymer network in which locally constrained crown motifs, polymer chain packing, and cooperative interactions collectively define the effective transport pathways. We have clarified this point in the revised manuscript.

Added text to methods section: In the prepared ultrathin films, selective complexation with K^+ ions is anticipated. Crown ethers free in solution can adapt their conformation to optimize coordination with a given ion; however, when incorporated into a densely crosslinked polyamide network, this flexibility is constrained, and the idealized cavity geometry may be distorted. As a result, ion coordination within the membrane should not be viewed as occurring through discrete pores of fixed size, but rather through locally constrained coordination environments defined by the crown ether moieties and the surrounding polymer matrix. These constrained sites likely exhibit reduced selectivity

compared to free crown ethers in solution, consistent with prior observations of crown ethers under restricted conformational freedom⁴².

2. Water contact angle data is critical for understanding membrane permeation behaviour in aqueous ion separation applications.

R: We thank the reviewer for this suggestion. Water contact angle measurements were performed and are now reported in the main text to complement the surface morphology characterization and to provide context for aqueous permeation behavior. Specifically, the sentence below was added.

Added text to the results section: This approach consistently produced smooth films with a low surface roughness of only 2.4 nm (Fig. 2E) and a moderately hydrophilic surface, as indicated by a water contact angle of $67 \pm 3^\circ$.

3. The nitration of dibenzo-18-crown-6 should yield both cis- and trans-DAB18C6. The methodology for their separation and the rationale for using only trans-DAB18C6 in this study should be detailed.

R: We thank the reviewer for this excellent comment and fully agree that polyamide films prepared from cis- and trans-DAB18C6 are expected to exhibit different structural and transport properties. Differences between cis and trans monomers are well known to influence network connectivity, chain packing, free volume, and transport behavior in polyamide systems, and similar effects are anticipated for crown ether-based polyamides.

As noted by the reviewer, nitration of dibenzo-18-crown-6 yields both cis and trans isomers. In this work, we followed well-established synthetic and separation procedures reported in the literature, in which the two regioisomers are separated based on their different solubilities during recrystallization from acetic acid, a step that selectively isolates the trans isomer with high purity. We have now explicitly specified this purification step in the manuscript and cited the appropriate literature to clarify how the trans isomer was obtained.

The primary objective of this study was to demonstrate that the selective complexation inherent to crown ethers can be translated into selective ion transport in an ultrathin polymeric membrane, rather than to systematically explore how monomer stereochemistry tunes membrane microstructure. For this reason, we focused exclusively on a single, well-defined isomer. We fully agree that comparing cis and trans crown ether monomers represents an important opportunity to further tune membrane structure and transport properties, and we have highlighted this direction in the Discussion section of the revised manuscript to guide future studies.

Added text to methods section: during which nitration of the aromatic rings produced a mixture of cis- and trans-di(nitrobenzo)-18-crown-6 isomers. Owing to its significantly lower solubility in the reaction medium, the trans isomer selectively precipitated from solution³⁵.

Added to the discussion section: In addition, the use of different crown ether regioisomers (e.g., cis versus trans) offers an unexplored handle to tune network topology, local packing, and ion–ligand accessibility in polymeric membranes. Likewise, varying the structure of the acid chloride comonomer, such as its rigidity or length, could provide a means to systematically control the spacing and packing density of crown ether units within the polyamide network. Together, these approaches offer complementary strategies to engineer crown ether-based membranes with higher degrees of molecular order and, consequently, enhanced ion selectivity and ion transport, while preserving the practical advantages of interfacial polymerization.

4. Have you tried the same polyamide membranes from cis- DAB18C6? How were the results?

R: We did not fabricate membranes using cis-DAB18C6 in this study. Exploring the use of cis versus trans crown ether monomers as a strategy to tune membrane microstructure, crosslinking topology, and transport selectivity is an important and interesting direction, but we believe it falls outside the scope of the present work. We have therefore added this point to the Discussion section of the manuscript to explicitly highlight monomer stereochemistry as a promising design parameter for future studies.

5. The polyelectrolyte gutter layer is essential in this study for preventing membrane cracks and leakage. Additional characterisation, including thickness and water and ion permeability, is necessary. Why was a 1.5-layer coating of PEI and PAA selected?

R: We agree with the reviewer that characterization of the polyelectrolyte gutter layer is important to decouple its contribution from the observed membrane performance. We therefore direct the reviewer to Figure S7 in the Supporting Information, which reports the ion transport properties of the gutter layer alone. These measurements show that the gutter layer exhibits high ion permeability and minimal ion–ion selectivity, in stark contrast to the crown ether–based polyamide selective layer. This striking difference demonstrates that the observed ion selectivity in the composite membranes originates from the ultrathin crown ether polyamide layer, while the gutter layer primarily serves a mechanical stabilization role.

The dry thickness of the gutter layer, estimated by depositing it directly onto a smooth Si wafer and measuring the resulting film, was below 2 nm, as reported in our previous manuscript^[2]. We note, however, that this value should be regarded as an approximation, as the actual thickness on AAO supports is expected to differ due to the dependence of layer-by-layer deposition on substrate properties such as surface charge and surface roughness. Using planar Si wafers to approximate the dry thickness of ultrathin layer-by-layer films is a common practice in the polyelectrolyte literature, but it represents a known limitation.

The choice of a 1.5-layer PEI/PAA coating reflects a balance between providing a smooth and chemically compatible surface for deposition of the ultrathin selective layer, while minimizing additional transport resistance. Similar to the role of PDMS gutter layers widely used in gas separation membranes due to their high gas permeance and compatibility with a wide variety of substrates and selective layers, here we sought a gutter layer that offers minimal resistance to the species of interest, namely ions. Polyelectrolyte multilayers are well suited for this role because they can be deposited conformally on rough supports, strongly adhere to both the substrate and the selective layer, and allow precise thickness control through the number of deposited layers.

Among polyelectrolyte systems, PEI/PAA is one of the most widely studied and best understood, with well-established deposition protocols and predictable growth behavior, which reduces uncertainty in membrane fabrication. As a result, the gutter layer does not impose a significant barrier to ion transport and does not contribute to ion selectivity, confirming that the observed separation performance originates from the crown ether–based polyamide layer.

6. The crown-ether polyamide layer demonstrated stability on PVDF porous support (Fig S2d). Was a gutter layer also applied to this sample?

R: We thank the reviewer for this comment. The SEM image shown in the Supporting Information for the crown ether–based polyamide layer on a PVDF porous support (Fig. S2D) does not include a gutter layer. However, similar delamination issues to those observed in the AAO-supported crown ether polyamide system were also observed for the PVDF-supported composite films upon immersing them in water. This indicates that, if transport studies were to be conducted using a PVDF support, the inclusion of a gutter layer would likewise be necessary to ensure stability during testing.

7. The mechanical properties of the selective layer are critical for membrane separation applications, despite minimised hydraulic pressure on ion separations. Can the membrane withstand pressure on an AAO substrate?

R: We thank the reviewer for raising this important point. We attempted to assess the mechanical robustness of the crown ether–based selective layers after transfer onto AAO substrates under applied transmembrane pressure. Owing to the extreme thinness of the films (~6 nm) and the rigidity and surface roughness of the AAO support, the transferred membranes could not withstand pressure-driven testing.

This limitation, together with the reviewer’s other comments, motivated us to explore direct fabrication of the selective layer on polymeric porous supports, where mechanical compliance and interfacial adhesion are more favorable. We have revised the manuscript accordingly to clarify the mechanical limitations of free-standing films on AAO supports and to avoid overstating their pressure stability, and we now discuss alternative fabrication strategies that are better suited for pressure-driven membrane applications.

Text modified in the abstract: Despite their minimal thickness, the flexible films with a structure formed by randomly arranged crown ether units could be handled and transferred for structural and transport characterization.

Text added to the results section: AAO supports were used as a rigid, well-defined model substrate to enable transport measurements of the ultrathin selective layer; they are not intended to represent pressure-bearing supports for practical membrane operation.

Note: Results for the newly fabricated crown ether-based membranes on PAN supports were added to the manuscript and are described in detail in the response to the final comment.

8. How does membrane performance vary with thickness? A thicker layer might decrease permeance but enhance selectivity.

R: We thank the reviewer for this comment. In this system, the self-limiting nature of interfacial polymerization, as evidenced in Figure 2b where increasing reaction time does not lead to thicker films, prevented us from fabricating selective layers with systematically varied thicknesses while preserving comparable microstructure. Although thicker films could, in principle, be obtained by altering monomer concentrations, introducing additives, or changing the solvents, such changes are well known to simultaneously affect cross-linking density, free volume, and chemical heterogeneity of polyamide films. As a result, any observed changes in permeance or selectivity would be convoluted with microstructural effects rather than reflecting thickness alone.

More broadly, as shown in our prior work^[3], the effect of thickness on selectivity in binding-mediated membranes could depend on whether ion transport is limited by entry into the membrane or by diffusion through the membrane via repeated complexation–decomplexation events. In that study, the thickness of layer-by-layer membranes containing coordination sites could be systematically varied. At high densities of coordination sites, weakly binding ions are largely excluded at the membrane–solution interface, making their permeance relatively insensitive to thickness. In contrast, ions that bind more strongly can enter the membrane and permeate through successive binding and release steps, rendering their transport increasingly diffusion-limited as thickness increases. As a result, increasing thickness could preferentially penalize the flux of the selectively bound ion, which could reduce (rather than enhance) selectivity. This framework helps explain why simply increasing membrane thickness does not necessarily improve separation performance and why many micrometer-thick membranes incorporating ion-complexing groups fail to exhibit meaningful selectivity. It also reinforces the rationale for focusing on ultrathin selective layers, where binding-mediated discrimination can be exploited without imposing excessive diffusive resistance.

9. Ion permeability performance along with selectivity should be compared with existing literature data.

R: We thank the reviewer for this comment. Although achieving the highest membrane performance was not the primary objective of this study, we agree that comparison with existing literature is important for benchmarking, particularly given the scattered nature of reported ion permeability and selectivity data. We have therefore added a comparison with relevant literature in the Supporting Information and cited it in the main text.

Added text to SI: Tables S5: Comparison of K^+/X^+ selectivity (where X^+ denotes other monovalent cations) and K^+ flux for the ultrathin crown ether-based polyamide membrane and reported membranes.

Added text to the results section: To place the reduced ion flux in context, we compared the K^+ flux and selectivity of the present membrane with those of reported K^+ -selective membranes (Table S5). While the K^+ flux of the crown ether-based polyamide is lower than that of many reported systems, it falls within a range expected for polymeric membranes that rely on binding-mediated selectivity rather than continuous ion-conducting pathways. The reduced flux reflects transport through a dense, highly crosslinked polyamide network with a non-ordered distribution of crown ether binding sites, some of which may not be accessible during ion transport.

10. The current fabrication method is complex. Given that interfacial polymerisation (IP) for RO membranes is directly applied to substrates for simplicity, can this crown ether polyamide layer also be directly applied? If so, how does it affect separation performance

R: We thank the reviewer for raising this important point. For a fundamental investigation focused on ion–ion selectivity and binding-mediated transport, we intentionally fabricated the selective layers at a free interface. This

approach enables clean characterization of the intrinsic chemistry, thickness, and morphology of the ultrathin selective layer using AFM, TEM, XPS, and QCM, and allows ion transport to be examined under diffusion-dominated conditions that exclude convective flow and minimize concentration polarization effects. Such conditions are critical for isolating ion–membrane interactions and mechanistic selectivity, which can be obscured under pressure-driven operation.

Motivated by the reviewer’s comment, we also explored direct interfacial polymerization of the crown ether polyamide layer on polymeric porous supports, analogous to conventional RO membrane fabrication. As widely documented, the properties of the support (surface chemistry, pore structure, and monomer uptake) strongly influence interfacial polymerization, and thus the resulting selective layer is not expected to be identical to films formed at a free interface. Using identical reaction conditions, we obtained continuous, mechanically robust membranes that could be evaluated under applied pressure.

Under pressure-driven stirred-cell testing, these membranes exhibited water permeances comparable to reported polyamide membranes and high rejection (>95%) of Rose Bengal, confirming complete coverage and the absence of macroscopic defects. However, no measurable selectivity between K^+ and other monovalent ions was observed. Under these conditions, convective solvent transport and concentration polarization can dominate ion transport and obscure binding-mediated selectivity mechanisms, particularly for ions of similar size and charge.

These results demonstrate that while direct fabrication on porous supports is feasible and yields mechanically robust membranes suitable for pressure-driven operation, the manifestation of ion selectivity depends strongly on the transport regime. We have added this discussion to the manuscript to clarify the distinction between intrinsic material selectivity and performance under pressure-driven operation.

Added text to the results section: To assess the behavior of the crown ether–based selective layer under pressure-driven conditions, membranes were also fabricated directly on porous polyacrylonitrile polymeric supports using identical interfacial polymerization conditions. Water permeance and salt rejection measurements were performed in a stirred cell at 3.5 bar, and high rejection of Rose Bengal (>95%) was observed, confirming complete surface coverage and the absence of macroscopic defects in the selective layer. As summarized in Table S4, these membranes exhibit water permeances of approximately $1.3 \text{ L m}^{-2} \text{ h}^{-1} \text{ bar}^{-1}$, comparable to conventional polyamide membranes, and a moderate rejection of Mg^{2+} (~32%), while showing low rejection (8–10%) for monovalent K^+ and Li^+ ions under both single-salt and mixed-salt conditions. The pressure-driven transport behavior observed here is consistent with that of conventional nanofiltration-type polyamide membranes, including those reported by Alhazmi et al., who employed the same amino-functionalized crown ether monomer in interfacial polymerization on polymeric supports and optimized fabrication conditions to leverage the intrinsic macrocyclic cavity for pharmaceutical separations³¹. These results indicate that under pressure-driven operation, convective solvent transport can dominate ion transport, limiting discrimination among ions of similar size and charge.

Added text to the discussion section: In this context, more stringent control over free volume distribution and crown ether alignment, such that continuous water pathways allowing ions to be transported together with water molecules are suppressed, may enable binding mediated ion selectivity to persist under pressure driven conditions.

Added text to the SI: Table S4.

Reviewer #2 (Remarks to the Author):

In this manuscript, the authors developed an ultra-thin, crown ether functionalized membrane using interfacial polymerization. Due to the specific interaction between 18-crown-6 and K^+ ions, along with the ultra-low membrane thickness of 6.6 nm, the membrane shows good selectivity between monovalent cations of 3.8 for K^+/Li^+ and 4.4 for K^+/Cs^+ . The proposed mechanism involves reversible complexation between the crown ether sites and K^+ ions. Overall, the work presents an interesting concept that could appeal to the membrane community. However, there are still some major issues that need to be addressed before considering its publication in Nature Communications.

R: We thank the reviewer for their careful evaluation and constructive feedback. We appreciate their recognition of the conceptual interest of our work and its relevance to the membrane community, and we have addressed the identified concerns in detail to strengthen the manuscript accordingly.

1. Figure 2h shows a rather low signal-to-noise ratio, which makes it difficult to clearly support the presence of unreacted $-NH_2$ groups. Additionally, the authors should consider discussing the potential impact of hydrolyzed $-COOH$ or protonated $-NH_3^+$ groups (under low pH conditions) on ion flux and selectivity, as these groups could facilitate the transport of anions or cations.

R: We thank the reviewer for their comments. In Figure 2h, the N1s spectrum was deconvoluted into amine- and amide-related components to justify that the presence of $-NH_2$ would be negligible if there was any. As the reviewer correctly pointed out, we believe the presence of unreacted $-NH_2$ was minimal, which is consistent with our statement in the original manuscript. For improved clarity, we have revised the statement in the original manuscript as follows:

“The N1s core-level spectrum provides additional evidence of successful crosslinking of the crown ether monomers, showing a single dominant nitrogen environment corresponding to amide nitrogen formed by reaction of acyl chloride with amine groups (Fig. 2H).”

Based on the XPS analysis, the primary charged functional groups under experimental conditions were $-COO^-$. It is noteworthy that under experimental conditions, ionization of $-COOH$ happened to a very limited extent. Our previous work^[2] has demonstrated that the role of $-COO^-$ in selective ion transport is complicated, involving four-body interactions among cations, anions, water molecules, and functional groups across bulk and confined environment at the partitioning and diffusion step, respectively. Therefore, we avoid over-interpreting the specific contribution of $-COO^-$. However, we indeed confirmed the essential role of crown ethers in achieving a high ion-ion selectivity through low-pH QCM and XPS ion adsorption experiments (Figure 3).

2. In Figure 3d, the author used XPS to quantify the adsorption of different ions. However, since XPS only probes the top few tens of nanometers, it mainly reflects surface information. It's strongly recommended to use ICP or other bulk characterization methods to support this analysis.

R: We thank the reviewer for this comment. Although XPS is generally regarded as a surface-sensitive technique, this limitation is not relevant in our case because the selective layers are ultrathin (~ 6 nm). To ensure that the entire film is probed and to calibrate the measurements, XPS samples were prepared on Au-coated glass substrates. The clear presence of the Au signal in the XPS spectra confirms (Figure R1) that the full thickness of the selective layer lies within the XPS sampling depth and also allows the Au peak to be used for energy calibration.

Figure R1. XPS of a ~6nm-thick crownether-based polyamide film atop a gold substrate.

3. The transfer process after interfacial polymerization introduces defects and hinder scalability. Could direct polymerization on a porous membrane (polymeric or ceramic) achieve comparable selectivity? In addition, on Page 9, line 22, the author mentions testing three “high-quality” samples for selectivity. How were these samples selected? Were they the top-performing membranes, or was there another criterion?

R: We thank the reviewer for raising these important points. For this fundamental study, the selective films were prepared at a free interface to enable clean characterization of the ultrathin selective layer without interference from the support. This approach allows direct correlation between the intrinsic chemistry, thickness, and morphology of the crown ether-based polyamide layer and its transport behavior using AFM, TEM, XPS, and QCM, and enables diffusion-dominated measurements that isolate binding-mediated ion selectivity.

Motivated by this comment, and in parallel with a related suggestion from Reviewer #1, we also explored translating the fabrication to direct interfacial polymerization on porous polymeric supports, analogous to conventional RO membrane fabrication. As widely documented for interfacial polymerization, the porous support influences the reaction through surface chemistry, pore structure, and monomer uptake; consequently, the resulting selective layer is not expected to be identical to films formed at a free interface. Using identical reaction conditions, we obtained continuous polyamide layers that were mechanically robust and suitable for pressure-driven testing. Under pressure-driven operation, these membranes exhibited water permeances comparable to reported polyamide membranes and showed high rejection (>95%) of Rose Bengal, confirming complete surface coverage and the absence of macroscopic defects. However, no measurable selectivity between K^+ and other monovalent ions was observed under these conditions. Under these conditions, convective solvent transport and concentration polarization can dominate ion transport and obscure binding-mediated selectivity mechanisms, particularly for ions of similar size and charge. These results demonstrate that while direct fabrication on porous supports is feasible and yields mechanically robust membranes suitable for pressure-driven operation, the manifestation of ion selectivity depends strongly on the transport regime. We have added this discussion to the manuscript to clarify the distinction between intrinsic material selectivity and performance under pressure-driven operation.

Regarding sample selection, the membranes used for selectivity analysis were not chosen based on peak performance. Instead, an objective integrity-based criterion was applied. Any membrane exhibiting a K^+ flux at least one order of magnitude lower than that of the bare polyelectrolyte-coated AAO support in the corresponding mixed-salt experiments was classified as “high-quality” and retained for analysis. This criterion ensures that measured transport reflects an intact crown ether-based selective layer rather than defects or pinholes. All membranes meeting this criterion were included, and the complete raw transport data have now been added to the Supporting Information (Tables S1–S3).

Added text to the results section: To assess the behavior of the crown ether-based selective layer under pressure-driven conditions, membranes were also fabricated directly on porous polyacrylonitrile polymeric supports using identical interfacial polymerization conditions. Water permeance and salt rejection measurements were performed in a stirred cell at 3.5 bar, and high rejection of Rose Bengal (>95%) was observed, confirming complete surface coverage and the absence of macroscopic defects in the selective layer. As summarized in Table S4, these membranes exhibit

water permeances of approximately $1.3 \text{ L m}^{-2} \text{ h}^{-1} \text{ bar}^{-1}$, comparable to conventional polyamide membranes, and a moderate rejection of Mg^{2+} (~32%), while showing low rejection (8–10%) for monovalent K^{+} and Li^{+} ions under both single-salt and mixed-salt conditions. The pressure-driven transport behavior observed here is consistent with that of conventional nanofiltration-type polyamide membranes, including those reported by Alhazmi et al., who employed the same amino-functionalized crown ether monomer in interfacial polymerization on polymeric supports and optimized fabrication conditions to leverage the intrinsic macrocyclic cavity for pharmaceutical separations³¹. These results indicate that under pressure-driven operation, convective solvent transport can dominate ion transport, limiting discrimination among ions of similar size and charge.

Added text to the results section #2: The full transport data for all measured membranes, including those not meeting the integrity criterion, are provided in the Supporting Information (Tables S1–S3).

Added text to the discussion section: In this context, more stringent control over free volume distribution and crown ether alignment, such that continuous water pathways allowing ions to be transported together with water molecules are suppressed, may enable binding mediated ion selectivity to persist under pressure driven conditions.

Added text to SI: Tables S1 to S4

4. The author mentions the membrane's mechanical robustness. Could this claim be supported up by experimental data, like AFM measurements?

R: We thank the reviewer for this comment, which helped us clarify the scope of the mechanical robustness claims in the manuscript. In the revised version, we removed ambiguous statements regarding mechanical stability and directly evaluated whether the ultrathin crown ether-based films transferred onto AAO supports could operate under applied hydraulic pressure. These films were not able to withstand pressure-driven testing, reflecting the extreme thinness of the selective layer and the mechanical mismatch with the rigid AAO support.

We note that crown ether-based polyamide films fabricated directly on polymeric porous supports were mechanically robust and could be tested under applied pressure. However, under pressure-driven operation these membranes did not exhibit measurable ion-ion selectivity, indicating that convective water transport can dominate ion transport and obscure binding-mediated selectivity. In addition, selective layers formed directly on porous supports are inherently different from those prepared at a free interface, as interfacial polymerization is strongly influenced by support properties such as surface chemistry, pore structure, and monomer uptake.

Accordingly, we have revised the manuscript to avoid overstating mechanical robustness and to clearly distinguish between structural integrity under handling, mechanical stability under pressure, and the transport regimes under which ion selectivity is observed.

Text modified in the abstract: Despite their minimal thickness, the flexible films with a structure formed by randomly arranged crown ether units could be handled and transferred for structural and transport characterization.

Text added to the results section: AAO supports were used as a rigid, well-defined model substrate to enable transport measurements of the ultrathin selective layer; they are not intended to represent pressure-bearing supports for practical membrane operation.

5. The ion absorption tests were done at low pH, were the selectivity measurements performed under the same acidic conditions? Also, would this low pH cause amide group hydrolysis and decrease long-term stability?

R: We thank the reviewer for pointing out that the pH used in the permeation experiments was not clearly stated. All permeation and selectivity measurements were performed at pH 6, and this information has now been added to the manuscript.

The ion sorption experiments were conducted at a lower pH by design. Under acidic conditions, electrostatic contributions to ion sorption arising from ionizable carboxylic acid groups in the polyamide are minimized. This allowed us to better isolate and assess ion sorption originating specifically from crown ether coordination rather than from electrostatic interactions associated with other functional groups in the polymer matrix.

Regarding stability, amide bond hydrolysis occurs at appreciable rates only under much more extreme acidic or basic conditions than those employed here. In addition, the ion sorption experiments were short in duration (less than two days). Under these conditions, hydrolysis of the polyamide backbone is not expected, and we therefore do not anticipate any impact on the measured ion sorption or selectivity.

Text modified in the the results section: In each experiment, the feed solution contained a binary salt mixture with equimolar amounts of KCl and the chloride salt of a competing cation at pH 6, while the receiving side was filled with deionized water (Fig. 4A).

6. Even though the membrane is very thin, the flux dropped by 1-2 orders of magnitude compared to the bare AAO substrate, this might be because it lacks ion-conducting groups. How does this flux performance compare to other ion-selective membranes? Also, could the authors suggest possible ways to improve the flux without compromising selectivity?

R: We thank the reviewer for this important comment. We acknowledge that, despite the ultrathin nature of the selective layer, the K^+ flux of the DB18C6-based polyamide membrane is substantially reduced relative to the bare AAO substrate.

To place this observation in context, we have included in the revised manuscript a comparison of K^+/X^+ selectivity and K^+ flux with reported ion-selective membranes (Table S5). The reduced flux reflects transport through a dense, highly crosslinked polyamide network with a non-ordered array of crown ether binding sites, some of which may not be accessible during transport, rather than through continuous ion-conducting pathways. This comparison highlights the well-known tradeoff between binding-mediated ion selectivity and permeability.

We also discuss strategies to improve ion flux without compromising selectivity, including promoting crown ether alignment, controlling free volume distribution, and tuning network topology through monomer design, as outlined in the revised Discussion section.

Added text to SI: Tables S5: Comparison of K^+/X^+ selectivity (where X^+ denotes other monovalent cations) and K^+ flux for the ultrathin crown ether-based polyamide membrane and reported membranes.

Added text to the the results section: To place the reduced ion flux in context, we compared the K^+ flux and selectivity of the present membrane with those of reported K^+ -selective membranes (Table S5). While the K^+ flux of the crown ether-based polyamide is lower than that of many reported systems, it falls within a range expected for polymeric membranes that rely on binding-mediated selectivity rather than continuous ion-conducting pathways. The reduced flux reflects transport through a dense, highly crosslinked polyamide network with a non-ordered distribution of crown ether binding sites, some of which may not be accessible during ion transport.

Modified text in the discussion section: While the monovalent ion selectivity achieved by the prepared membranes is impressive for a polymeric system, it falls short of the higher selectivities observed in highly ordered crystalline channels such as crown ether-functionalized MOFs²², where the spatial arrangement of crown ether molecules is precisely controlled. However, translating these crystalline channels into continuous membranes remains challenging, as ion transport in polycrystalline systems is typically dominated by grain boundaries and defects rather than by the intrinsic selectivity of the channels. In contrast, polymeric membranes are readily scalable, which motivates efforts to enhance their selectivity and transport rates to approach those of crystalline systems. Bridging the performance gap between highly ordered but difficult-to-scale crystalline systems and polymeric membranes fabricated via scalable interfacial polymerization could be achieved by aligning crown ether molecules during polymerization. Recent advances with cyclodextrin-based building blocks have demonstrated that functionalizing them to promote their alignment at the water/organic interface during interfacial polymerization can produce crosslinked films of aligned macrocycle molecules with significantly improved solute-solute selectivities compared to disordered films³³. Applying similar alignment strategies to crown ether building blocks could enhance the molecular ordering in the resulting membranes, narrowing the selectivity gap while preserving the scalability and processability that make polymeric systems so attractive. In this context, more stringent control over free volume distribution and crown ether alignment, such that continuous water pathways allowing ions to be transported together with water molecules are suppressed, may enable binding mediated ion selectivity to persist under pressure driven conditions.

In addition, the use of different crown ether regioisomers offers an unexplored handle to tune network topology, local packing, and ion–ligand accessibility in polymeric membranes. Likewise, varying the structure of the acid chloride comonomer, such as its rigidity or length, could provide a means to systematically control the spacing and packing density of crown ether units within the polyamide network. Together, these approaches offer complementary strategies to engineer crown ether–based membranes with higher degrees of molecular order and, consequently, enhanced ion selectivity and ion transport, while preserving the practical advantages of interfacial polymerization.

7. Given the extensive literature on the use of crown ethers in interfacial polymerization for ion-selective membranes [Angewandte Chemie, 2023, 62(18): e202300167; Journal of Materials Chemistry A, 2025, 13(16): 11732-11748; Journal of Membrane Science, 2025, 714: 123372], the novelty and uniqueness of this study should be further clarified and emphasized.

R: We thank the reviewer for highlighting the growing body of literature on crown ether–based membranes fabricated via interfacial polymerization. We have revised the manuscript to more clearly position the present work within this context and to clarify its distinct contribution.

While recent studies have successfully incorporated crown ether and other macrocyclic monomers into IP-derived thin films, these efforts have primarily leveraged macrocycles as rigid, bulky building blocks that introduce sub-nanometer free volume to enhance solute transport or solute–solute selectivity, targeting monovalent–divalent ion separations, proton conduction, or organic solute purification. In contrast, the present study demonstrates a fundamentally different transport mechanism, in which the ion-coordination chemistry of crown ethers, rather than their role as static pore formers, governs selective transport between closely related monovalent ions.

Specifically, our work reveals a binding-mediated transport regime in which reversible ion coordination within a densely crosslinked polymer matrix biases ion–ion selectivity even in the absence of well-defined crystalline channels. This distinction, together with the use of ultrathin films and diffusion-dominated transport measurements to isolate intrinsic ion–crown ether interactions, differentiates the present study from prior IP-based crown ether membranes. We have emphasized this point in the revised manuscript and added the references suggested by the reviewer to clearly delineate how our work advances beyond existing approaches.

Text added to the results section: Despite these challenges, several recent studies have demonstrated the feasibility of incorporating crown ether–based monomers into polymeric thin films via interfacial polymerization, primarily targeting separations such as monovalent–divalent ion pairs²⁹, proton conduction pathways³⁰, or pharmaceutical purifications³¹. In these studies, crown ether moieties are largely exploited as macrocyclic cavities that introduce well-defined sub-nanometer free volume absent in analogous linear or non-macrocyclic polyether monomers, thereby creating additional and more uniform transport pathways that enhance the transport of only certain molecules. Related work has also shown that other bulky macrocycles, including cyclodextrins and trianglamines, can be successfully integrated into ultrathin IP-derived membranes, where their intrinsic porosity similarly enhances solute–solute selectivity³²⁻³⁴. Here, we show that the ion-coordination properties of crown ethers can also be harnessed to achieve selective transport between closely related monovalent ions, revealing a distinct transport regime in which selective complexation governs ion–ion selectivity.

Reviewer #3 (Remarks to the Author):

The paper describes the preparation and characterization of interfacial polymerized thin-film composite membranes with an amino-functionalized crown ether as hydrophilic monomer. The key finding of the work is the preferential K^+ transport over other monovalent cations. The ion-ion selectivities of approximately 4 for monovalent cation pairs (K^+/Cs^+ and K^+/Li^+) is impressive. The conclusions are supported by the well-described experiments.

R: We thank the reviewer for their positive and thoughtful assessment. We appreciate their recognition of the significance of the observed K^+ selectivity and their acknowledgment that the conclusions are well supported by the experimental results

1. The paper would have more relevance for applications if the IP films would have been prepared on a polymeric support membrane. Here the support was an anodic aluminium oxide membrane. These membranes are brittle and prevent applications under any mechanical stress like pressure filtration. In addition they are not compatible with the polymeric IP film. This required a polyelectrolyte inter layer which influences the cation transport. The ions transport properties were not measured under filtration condition but in a diffusion cell, which has little relevance for practical applications.

R: We thank the reviewer for this thoughtful and detailed comment. We agree that preparing interfacial polymerized films directly on polymeric supports and evaluating them under pressure-driven conditions is essential for assessing application-oriented membrane performance. In the present work, however, our primary objective was to establish and interrogate the intrinsic ion-ion selectivity arising from crown ether coordination within an ultrathin polymeric selective layer.

To this end, the selective films were intentionally prepared at a free interface and transferred onto AAO supports. This approach enabled clean structural and chemical characterization of the ultrathin selective layer using AFM, TEM, XPS, and QCM, and allowed ion transport to be examined under diffusion-dominated conditions that minimize convective flow and concentration polarization. These conditions are critical for isolating binding-mediated ion selectivity, which can be difficult to resolve under pressure-driven operation. The AAO support was used as a rigid, well-defined model substrate to enable transport measurements and is not intended to represent a pressure-bearing support for practical applications.

Motivated by this comment, and to directly assess the feasibility of translating the fabrication to application-relevant architectures, we also explored direct interfacial polymerization of the crown ether-based selective layer on polymeric porous supports, analogous to conventional RO membrane fabrication. As widely documented, the properties of the porous support strongly influence interfacial polymerization through surface chemistry, pore structure, and monomer uptake; consequently, the resulting selective layer is not expected to be identical to films formed at a free interface. Using identical reaction conditions, we obtained continuous polyamide layers that were mechanically robust and suitable for pressure-driven testing.

Under pressure-driven stirred-cell operation, these membranes exhibited water permeances comparable to reported polyamide membranes and showed high rejection (>95%) of Rose Bengal, confirming the absence of macroscopic defects. However, no measurable selectivity between K^+ and other monovalent ions was observed under these conditions, indicating that pressure-driven transport emphasizes solvent flow and can obscure binding-mediated ion discrimination for closely related ions.

Together, these results demonstrate that while direct fabrication on polymeric supports is feasible and yields mechanically robust membranes suitable for pressure-driven operation, the manifestation of ion selectivity depends strongly on the transport regime. We have revised the manuscript to clarify this distinction and to explicitly differentiate between intrinsic, binding-mediated selectivity probed under diffusion-dominated conditions and performance under pressure-driven operation. Additional pressure-driven transport data on polymeric supports have been added to the Results and Supporting Information (Table S4), and this discussion has been incorporated into the revised manuscript.

Added text to the results section: To assess the behavior of the crown ether–based selective layer under pressure-driven conditions, membranes were also fabricated directly on porous polyacrylonitrile polymeric supports using identical interfacial polymerization conditions. Water permeance and salt rejection measurements were performed in a stirred cell at 3.5 bar, and high rejection of Rose Bengal (>95%) was observed, confirming complete surface coverage and the absence of macroscopic defects in the selective layer. As summarized in Table S4, these membranes exhibit water permeances of approximately $1.3 \text{ L m}^{-2} \text{ h}^{-1} \text{ bar}^{-1}$, comparable to conventional polyamide membranes, and a moderate rejection of Mg^{2+} (~32%), while showing low rejection (8–10%) for monovalent K^+ and Li^+ ions under both single-salt and mixed-salt conditions. The pressure-driven transport behavior observed here is consistent with that of conventional nanofiltration-type polyamide membranes, including those reported by Alhazmi et al., who employed the same amino-functionalized crown ether monomer in interfacial polymerization on polymeric supports and optimized fabrication conditions to leverage the intrinsic macrocyclic cavity for pharmaceutical separations³¹. These results indicate that under pressure-driven operation, convective solvent transport can dominate ion transport, limiting discrimination among ions of similar size and charge.

Added text to the discussion section: In this context, more stringent control over free volume distribution and crown ether alignment, such that continuous water pathways allowing ions to be transported together with water molecules are suppressed, may enable binding mediated ion selectivity to persist under pressure driven conditions.

Added text to the SI: Table S4.

2. The authors should have cited the work of Alhazmi et al. (ref. 32) in a correct way. The membrane preparation part in this reference is very similar, also using interfacial polymerisation and amino-functionalized crown ether as hydrophilic monomer. But they use a polymeric support membrane and pressure filtration for transport characterisation. This should have been discussed in this work.

R: We thank the reviewer for pointing this out. The manuscript has been revised to correctly cite and explicitly discuss the work of Alhazmi et al. (ref. 31 in the updated manuscript).

Added text #1 to the results section: Despite these challenges, several recent studies have demonstrated the feasibility of incorporating crown ether–based monomers into polymeric thin films via interfacial polymerization, primarily targeting separations such as monovalent–divalent ion pairs²⁹, proton conduction pathways³⁰, or pharmaceutical purifications³¹. In these studies, crown ether moieties are largely exploited as macrocyclic cavities that introduce well-defined sub-nanometer free volume absent in analogous linear or non-macrocyclic polyether monomers, thereby creating additional and more uniform transport pathways that enhance the transport of only certain molecules.

Added text #2 to the results section: The pressure-driven transport behavior observed here is consistent with that of conventional nanofiltration-type polyamide membranes, including those reported by Alhazmi et al., who employed the same amino-functionalized crown ether monomer in interfacial polymerization on polymeric supports and optimized fabrication conditions to leverage the intrinsic macrocyclic cavity for pharmaceutical separations³¹.

3. The authors describe on page 5 that thin membranes favor the high selectivity because the rate limiting step for the transport of the non-favored ion is the partitioning into the membrane, whereas the rate limiting step for the target ion is its movement through the network of binding sites. For me it is not convincing that the solution of the not complexed ion into the water-swollen membrane is rate limiting.

R: We thank the reviewer for this insightful comment. Our intent was not to claim a definitive rate-limiting step, but rather to propose a plausible hypothesis for how ion–ion selectivity may arise in the ultrathin, binding-mediated crown ether–based membranes studied here.

Recent studies have identified ion partitioning at the membrane–solution interface as a key determinant of ion–ion selectivity in several polymeric membrane systems^[3–5]. Within this framework, the effect of membrane thickness on selectivity depends on whether ion transport is limited primarily by entry into the membrane or by diffusion through the membrane.

As demonstrated in our prior work^[3], thickness-controlled layer-by-layer membranes containing coordination sites provide a useful model to examine these effects. In those systems, weakly binding ions are largely excluded at the membrane–solution interface, rendering their permeance relatively insensitive to thickness. In contrast, more strongly

binding ions can enter the membrane and permeate through successive binding and release steps, such that their transport becomes increasingly diffusion-limited as membrane thickness increases. As a result, increasing thickness can preferentially penalize the flux of the selectively bound ion and reduce, rather than enhance, overall selectivity.

We hypothesize that a similar transport picture applies to the present ultrathin interfacially polymerized membranes, where selective coordination enhances the effective mobility of the target ion relative to competing ions. However, because interfacial polymerization does not allow independent control of membrane thickness without simultaneously altering membrane chemistry and internal structure, this mechanism cannot be directly verified through thickness-dependent measurements in the present system. We have therefore revised the manuscript to explicitly present this transport picture as a hypothesis and to clarify the language so that it is not interpreted as a demonstrated rate-limiting process.

Text modified in the results section: The thinness of crown ether-containing membranes is crucial when aiming to leverage the reversible and selective interactions of crown ether groups for ion transport. In such membranes, where selective transport relies on these reversible interactions, we hypothesize that the dominant transport resistance for the target ion arises from the movement through the network of binding sites^{4,36}. This process is governed by the repeated reversible binding and release of the ion as it traverses the membrane. A thicker membrane is therefore expected to slow this process, as the ion would need to undergo more interactions before crossing to the other side. Conversely, for competing ions that interact weakly with the binding sites, transport is not selectively facilitated and may be governed by interfacial partitioning into the membrane, such that increasing thickness has a comparatively smaller effect on their transport^{36,37}. As membrane thickness increases, the transport of the target ion—selectively binding with the functional groups—can be penalized more strongly than that of competing ions. We recently demonstrated this using a model system containing a high density of Cu²⁺-selective iminodiacetate groups, where membrane thickness could be precisely controlled and increasing thickness led to a decline in copper selectivity³⁶.

Response to Reviewers References:

- [1] Z. Liu, S. K. M. Nalluri, J. F. Stoddart, *Chem. Soc. Rev.* 2017, 46, 2459.
- [2] J. Zhang, L. F. Villalobos, J. Lee, M. Zhong, M. Elimelech, *ACS Appl. Mater. Interfaces* 2025, 17, 30817.
- [3] R. M. DuChanois, M. Heiranian, J. Yang, C. J. Porter, Q. Li, X. Zhang, R. Verduzco, M. Elimelech, *Science Advances* 2022, 8, 1.
- [4] R. S. Kingsbury, M. A. Baird, J. Zhang, H. D. Patel, M. J. Baran, B. A. Helms, E. M. V. Hoek, *Matter* 2024, 7, 2161.
- [5] Y. Guo, J. He, J. Zhang, M. Sheng, Z. Wang, M. Elimelech, L. Wang, *Science Advances* 2025, 11, eadu8302.